# Faecal microbiota of schoolchildren is associated with nutritional status and markers of inflammation: a double-blinded cluster-randomized controlled trial using multi-micronutrient fortified rice

Yohannes Seyoum[1], Valérie Greffeuille[1], Dorgeles Kouakou Dje Kouadio[1], Khov Kuong[2], Williams Turpin[1,5,6], Rachida M'Rabt[1], Vincent Chochois[1], Sonia Fortin[1], Marlène Perignon[1,7], Marion Fiorentino[1,8], Jacques Berger[1], Kurt Burja[3], Maiza Campos Ponce[4], Chhoun Chamnan[2], Frank T. Wieringa[1,9] & Christèle Humblot[1,9] ✉

Faecal microbiota plays a critical role in human health, but its relationship with nutritional status among schoolchildren remains under-explored. Here, in a double-blinded cluster-randomized controlled trial on 380 Cambodian schoolchildren, we characterize the impact of six months consumption of two types of rice fortified with different levels of vitamins and minerals on pre-specified outcomes. We investigate the association between the faecal microbiota (16SrRNA sequencing) and age, sex, nutritional status (under-weight, stunting), micronutrient status (iron, zinc and vitamin A deficiencies, anaemia, iron deficient anaemia, hemoglobinopathy), inflammation (systemic, gut), and parasitic infection. We show that the faecal microbiota is characterised by a surprisingly high proportion of *Lactobacillaceae*. We discover that deficiencies in specific micronutrients, such as iron and vitamin A, correlate with particular microbiota profiles, whereas zinc deficiency shows no such association. The nutritional intervention with the two rice treatments impacts both the composition and functions predicted from compositional analysis in different ways. (ClinicalTrials.gov (Identifier: NCT01706419))

The gut microbiota plays many crucial roles, ranging from digesting food nutrients not digested by human digestive enzymes, to having a strong influence on non-communicable diseases such as obesity[1]. While the gut microbial profile of young children (< 3 years of age) and of adults has been investigated repeatedly in different parts of the world, data on older children are more scarce[2]. It is generally acknowledged that the composition of the gut microbiota varies dramatically during the first year of life, before stabilising at around 2 to 3 years of age, when it is similar to that of adults[3]. However, some studies that examined the gut microbiota beyond childhood have shown that the gut microbiota diversity changes throughout the lifespan[2].

Different factors such as treatment with antibiotics and nutritional status strongly affect the gut microbiota, and diet is among the most important factors[4]. Panels of metadata describing host and dietary factors enabled identification of the external factors that

contribute to variations in gut microbiota[2]. However, studies on large population-based cohorts emphasised huge interindividual microbial diversity[2]. Studies of the gut microbial composition of individuals from less studied countries (i.e., non-Western countries) involving large numbers of children together with the collection of metadata panels are essential to better understand the relationships between bacterial profiles and environmental or host intrinsic factors.

Indeed, increasing numbers of investigators are testing the use of the gut microbial composition as a tool to predict and/or mitigate the risks of different human diseases[2]. Considering the impact of diet on gut microbiome, finding microbial biomarkers for nutritional status would advance our understanding of the role of bacteria in health and disease. Ultimately, once our understanding of diet-microbiome-nutritional status mechanisms has improved, it could provide a non-invasive way of fighting malnutrition by targeting the gut microbiota.

While a large body of literature reports associations between the human microbiome and obesity, even if the results are inconsistent[5], data on bacterial microbiota in children with acute malnutrition are relatively scarce but once again underline the importance of gut microbiota[6]. Indeed, severe and acute malnutrition is associated with significant relative microbiota immaturity[7,8]. The use of gnotobiotic mice also identified the gut microbiome as a causal factor in severe acute malnutrition, in addition to insufficient nutrient intake[9]. Significantly, the use of microbiota-targeted complementary food was more efficient than ready-to-use supplementary foods in increasing weight in moderately malnourished Bangladeshi children[10].

Moreover, in most low-and middle-income countries, other forms of malnutrition such as chronic malnutrition leading to stunting and micronutrient deficiencies remain highly prevalent, thereby raising other major health concerns (poor growth performance and increased risk of morbidity and mortality from infectious diseases). In Cambodia, malnutrition remains a major problem despite a considerable reduction in national poverty since the mid-1990s. In 2014, stunting and anaemia affected respectively, 32.4% and 55.0% of children under 5 years of age. Micronutrient deficiencies were highly prevalent with ≥ 80% of children found to be deficient in zinc[11].

The aim of the 'Fortified Rice for School Children in Cambodia' (FORISCA) project was to assess the impact of rice fortified with different levels of vitamins and minerals on nutritional status, development and anthropometry of 9500 schoolchildren aged 6–14 years, who received fortified rice or normal rice for six months. Anthropometry measurement showed that he prevalence of stunting was 40.0% at baseline[12]. The FORISCA trial showed that daily consumption of three types of fortified rice with different micronutrient content provided through a school meal programme improved the status of most micronutrients[13,14], had a small but nevertheless significant impact on cognitive development[15], but simultaneously increased the prevalence of hookworm infection[16]. Evidence for gut and systemic inflammation was found in respectively 5.4% and 39.5% of the children[17]. The three different formulations of fortified rice had different impacts on these parameters[14,16].

Numerous studies have investigated the role of gut microbiota in micronutrient status. Indeed, early work comparing germ-free and conventional animals showed a small but crucial role of the gut microbiota in the availability of different B-vitamins (B5, B8, B9, B12 but not B1) in animals fed a deficient diet[18]. Zinc requirements were also found to be reduced in the germ-free state, which was not the case for iron requirements[18]. Some evidence for associations between micronutrient status and specific gut microbiota composition has been reported, for example a decrease in the relative abundance of *Bifidobacterium* has been associated with iron supplementation[19,20].

Studying the faecal bacterial composition of underexplored populations such as schoolchildren in non-Western countries, and its relationship with nutritional status is crucial to advance existing knowledge. Furthermore, double-blind cluster randomized controlled trials that investigate the effect of nutritional intervention on faecal bacteria composition in a large number of subjects are rare in the literature[21].

In this work, we describe microbiota of the Cambodian schoolchildren participating in the FORISCA trial and we investigate associations between age, sex, (micro)nutrient status (underweight, stunting, anaemia and biomarkers of iron, zinc and vitamin A status), hemoglobinopathy, inflammation (systemic, gut), and parasitic infection. To this end, the bacterial composition of faecal samples collected from a subsample of 380 schoolchildren participating in the FORISCA project is analysed at the baseline. In addition, we measure the impact of consuming two types of fortified rice with differing nutritional composition, on the faecal microbiota of the children. We also predict the functionality of faecal microbiota from the composition analysis. Here we show that the gut microbiota of Cambodian school-aged children is characterised by a high proportion of *Lactobacillaceae*. Some bacterial features are associated with micronutrient status, anthropometry and inflammatory status. Consumption of multiple micronutrient-fortified rice at school for six months, modifies the children's bacterial composition as well as the predictive functional characteristics with different effects depending on the rice treatment.

## Results
### Baseline description of the population characteristics and the effect of the intervention on micronutrient status
This study involved a subset of 380 children randomly selected from the parent FORISCA double-blind cluster randomized controlled trial using multi-micronutrient fortified rice on 9500 children (ClinicalTrials.gov identifier NCT01706419). The characteristics of the participants at baseline are listed in Supplementary Table 1. The mean age of the children was 9.7 years (range 6–14 years old) and 53% of the children were boys. Twenty percent of the children were anaemic, and 45% of the children were stunted (Supplementary Table 1). Prevalence of iron and zinc deficiencies were 51% and 89%, respectively, but the prevalence of vitamin A deficiency was lower ( < 8%). Thirty-seven percent of the children showed evidence of systemic inflammation while only 3% of the children had gastrointestinal inflammation. Infection by parasites was detected in 27% of the children.

The 380 schoolchildren had been randomized (with school as cluster) to receive either normal rice (Placebo), or one of the two types of fortified rice: UltraRice Original formulation (UR-original) or UltraRice Improved formulation (UR-improved) for a period of six months. UR-original was fortified with four micronutrients (iron, zinc, vitamin B1 and B9). UR-improved had four additional vitamins (vitamin A, vitamins B3 and B12). Their final composition differed slightly, with UR-original providing slightly more iron and zinc than UR-improved. The intervention had no significant impact on stunting. (Supplementary Table 2).

The impact of the two interventions on micronutrient status differed, for details, see supplementary Table 3. Consumption of UR-original was only able to reduce the prevalence of zinc deficiency, whereas consumption of UR-improved had a wider effect, by reducing anaemia, vitamin A and zinc deficiency as well as inflammation.

### The faecal bacterial communities of Cambodian schoolchildren are characterised by a surprisingly high proportion of *Lactobacillaceae*
To investigate the faecal bacterial composition of the 380 schoolchildren, we applied 16 S rRNA amplicon sequencing at baseline and again after six months of nutritional intervention. The number of reads varied from 2948 to 314,227, with an average read count per sample of 39,355. Following sequencing, a total of 44,129,385 reads were generated, and after quality filtering, 67.7% of the sequences were retained, resulting in 29,910,230 reads from 760 faecal samples.

At baseline, the gut microbiota of Cambodian school-aged children was characterised by a large proportion of Firmicutes, followed

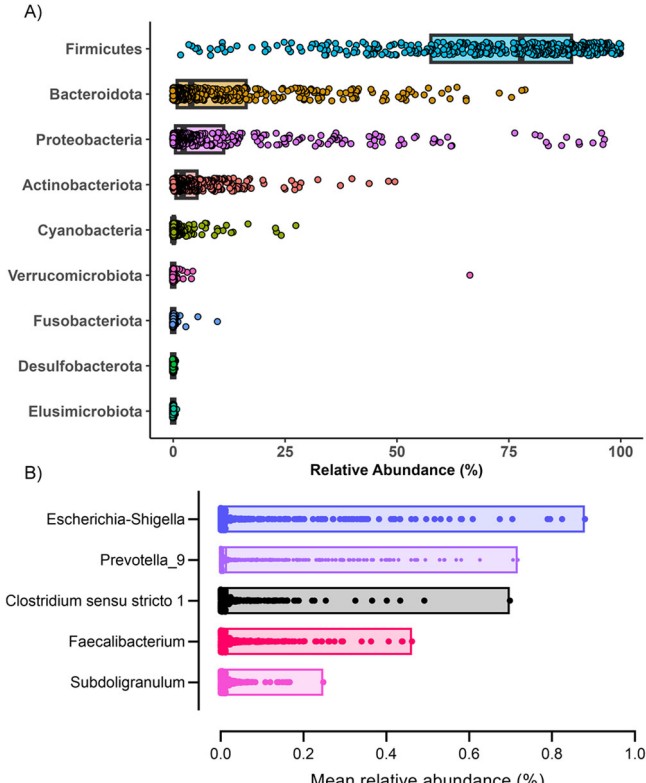

**Fig. 1 | Description of the faecal bacterial composition of the Cambodian schoolchildren children before nutritional intervention (*n* = 380). A** Relative abundance of bacterial phyla based on 16 S data. Box plot represents median and minimum and maximum. **B** Distribution of the abundances of the genera part of the core microbiota. The core microbiota is defined as the list of genera present in at least 95% of the samples in all the population. Box plot represents median and minimum and maximum. Source data are provided as a Source Data file.

by Bacteroidota (formerly termed Bacteroidetes), Proteobacteria and Actinobacteria (Fig. 1A). Firmicutes mainly comprised the class Bacilli (order Lactobacillales, family *Lactobacillaceae*, genus *Lactobacillus*) (Supplementary Fig. 1). The core microbiota in the children, i.e., genera shared by 95% of the individuals, comprised only five genera (Fig. 1B).

**Baseline faecal bacterial composition is linked with the micronutrient status and age of the schoolchildren**

Differences in microbiota among children grouped according to age, sex, nutritional, micronutrients, inflammatory status and parasitic infection were evaluated based on their alpha-diversity using Kruskal Wallis statistical analysis and on their beta-diversity using Permutational Multivariate Analysis of Variance (PERMANOVA) analysis (Table 1).

Among the 13 variables studied, sex, underweight, stunting, iron and zinc deficiencies, haemoglobinopathy (mostly haemoglobin E), inflammation and parasitic infection were not linked with the level of diversity within individuals (alpha-diversity, Supplementary Table 4). Pielou's evenness distribution differed significantly in children with or without anaemia, iron deficiency anaemia, vitamin A deficiency: children with deficiency had a lower diversity index than non-deficient children (Table 1). When the Shannon index was used, children with or without vitamin A deficiency exhibited different richness (number of species present in each sample) and evenness (species abundance) in the same direction.

Considering the age range of the study population, we decided to independently check if younger children (6–9 years old, *n* = 211) harboured different alpha-diversity than children in prepuberty or puberty (10–14 years old, *n* = 169). Both Pielou's evenness and Shannon indexes revealed differences: the older children had a higher diversity index than the younger ones (Table 1).

The level of diversity between children (beta-diversity) differing in their underweight, stunting, anaemia, iron deficiency anaemia, zinc, haemoglobinopathy, inflammation and parasitic infection status was not statistically different (Supplementary Table 5). Bray-Curtis (abundance of taxa) and Jaccard (presence/absence of taxa) distances differed significantly in children with (*n* = 193) or without (*n* = 187) iron deficiency (Table 1). Similarly, Bray-Curtis and weighted Unifrac (different phylogenetic lineages considering the abundance of taxa) distances differed significantly in children with (*n* = 30) or without (*n* = 350) vitamin A deficiency.

Furthermore, beta-diversity differed between age groups as shown by their Bray-Curtis, Jaccard and Unweighted Unifrac (phylogenetic lineages of taxa) distances (Table 1). Finally, Jaccard distances differed significantly in boys (*n* = 211) and girls (*n* = 169).

**Some bacterial features are significantly associated with nutritional, inflammatory and anthropometrical status**

We identified the bacterial genera most likely to explain differences between children of different age, sex, nutritional, micronutrients, inflammatory status and parasitic infection, using the Linear discriminate analysis Effect Size (LefSE) method at baseline. No associations between bacterial genera and iron deficiency, zinc deficiency, haemoglobinopathy and gut inflammation were identified.

While some bacterial groups were significantly associated with anaemia, regardless of the underlying cause, other bacterial groups were associated with iron deficiency anaemia (Fig. 2). Indeed, anaemia (*n* = 76) was associated with *Ruminococcus torques group*, *Anaerostipes* and *Prevotella* (Fig. 2A). In contrast, iron deficient anaemia (*n* = 47) was associated only with *Prevotella* (Fig. 2B). The bacterial profile of non-anaemic children (*n* = 304) was characterised by *Limosilactobacillus*, *Erysipelatoclostridiaceae* and *Klebsiella*. Children with no iron deficiency anaemia (n = 333) - however, this group may include children who are anaemic for reasons other than iron deficiency - were characterised by different bacterial associations, including Monoglobales or Oscillospirales. Some bacteria, such as *Clostridia*, *Lachnospiraceae*, *Faecalibacterium* and *Eubacterium halli* were associated with both non-anaemic status and non-iron deficient anaemia.

Vitamin A deficiency (*n* = 30) was associated with Bacilli (*Lactobacillaceae*), *Gordonibacter* and *Romboutsia* (Fig. 2C), whereas *Desulfobvibrionia*, Oscillospirales and Lachnospirales were abundant in children with sufficient vitamin A (*n* = 350) (Fig. 2C).

The microbiota of stunted children (*n* = 172) was characterised by *Prevotella_7* and *Holdemanella* (Fig. 2D). The faecal microbiota of non-stunted children (*n* = 208) was characterised by Clostridia, *Marinifilaceae*, *Succinivibrionaceae*, *Odoribacter*, *Alistipes*, and *Raoultibacter*.

Systemic inflammation (*n* = 140) was associated with Firmicutes (Supplementary Fig. 2A). In absence of inflammation (*n* = 240), there was an association with around 10 taxa. Infestation by parasites (*n* = 104) was associated with Desulfovibrionia, *Hungateiclostridiaceae*, *Erysipelotrichaceae*, *Weissella* and Sarcina (Supplementary Fig. 2B). The absence of parasitic infestation (*n* = 276) was associated with the genus *Kurthia*.

The microbiota of the group of younger children (6–9 years old) was significantly associated with Firmicutes (Bacilli, Lactobacillales, *Lactobacillus*), *Butyricoccaceae*, *Anaerococcus* and *Agathobacter*, while the microbiota of older children was associated with nine taxa, including Fusobacteriota, Negativicutes and Oscillospirales (Supplementary Fig. 2C). Finally, the microbiota of girls was characterised by Clostridiales and Staphylococcales while the microbiota of boys was characterised by Actinobacteria, Desulfobacterota, *Oscillospiraceae*, *Eubacterium*, *Coprococcus*, *Anaerovoraceae* and *Ligilactobacillus* (Supplementary Fig. 2D).

**Table 1 | Baseline alpha- and beta-diversity (pairwise distance) metrics of the 380 children grouped according to age, sex, nutritional micronutrient and inflammatory status and parasitic infection**

| Alpha diversity | | | | |
|---|---|---|---|---|
| | **Variable** | | **Kruskal-Wallis *H*** | ***q*-value** |
| Shannon index | Age | 6–9 years (*n* = 211, 4.6 ± 1.24) < 10–14 years (*n* = 169, 4.95 ± 1.12) | 6.333 | 0.0119 |
| | Vitamin A deficiency | yes (*n* = 30, 4.35 ± 1.13) < no (*n* = 350, 4.81 ± 1.2) | 4.938 | 0.0263 |
| Pielou's evenness | Age | 6–9 years (*n* = 211, 0.66 ± 0.15) < 10–14 years (*n* = 169, 0.69 ± 0.13) | 6.839 | 0.0089 |
| | Anaemia | yes (*n* = 76, 0.64 ± 0.17) < no (*n* = 304, 0.68 ± 0.13) | 4.726 | 0.0297 |
| | Iron deficiency anaemia | yes (*n* = 47, 0.63 ± 0.17) < no (*n* = 333, 0.68 ± 0.14) | 4.281 | 0.0385 |
| | Vitamin A deficiency | yes (*n* = 30, 0.61 ± 0.14) < no (*n* = 350, 0.68 ± 0.14) | 5.646 | 0.0175 |
| **Beta diversity** | | | | |
| | **Variable** | | ***Pseudo-F*** | ***q*-value** |
| Bray Curtis | Age | 6–9 years (*n* = 211) vs. 10–14 years (*n* = 169) | 2.319 | 0.0020 |
| | Iron deficiency | yes (*n* = 193) vs. no (*n* = 187) | 1.409 | 0.0410 |
| | Vitamin A deficiency | yes (*n* = 30) vs. no (*n* = 350) | 1.525 | 0.0230 |
| Weighted UniFrac | Vitamin A deficiency | yes (*n* = 30) vs. no (*n* = 350) | 2.540 | 0.0330 |
| Unweighted UniFrac | Age | 6–9 years (*n* = 211) vs. 10–14 years (*n* = 169) | 1.490 | 0.0440 |
| Jaccard | Age | 6–9 years (*n* = 211) vs. 10–14 years (*n* = 169) | 1.291 | 0.0070 |
| | Sex | boy (*n* = 200) vs. girl (*n* = 180) | 1.888 | 0.0350 |
| | Iron deficiency | yes (*n* = 193) vs. no (*n* = 187) | 1.215 | 0.0230 |

Differences in alpha-diversity were estimated using a two-way Kruskal Wallis test. The direction of the change is indicated by "<". Differences in beta-diversity were estimated using a two-way PERMANOVA analysis with 999 permutations. The number of samples in each subcategory is indicted in brackets. For all alpha diversity measures, the median ± standard deviation is presented alongside the number of participants (n, median± standard deviation). Only variables with statistically significant differences are listed in the table.

Age and sex were identified as potential confounding variables, as they were associated with differences in beta-diversity (Table 1) and in composition (Supplementary Fig. 2). Therefore, the significant features identified by LefSE analysis were integrated in a linear mixed model, with age groups (6−9 and 10−14 years old) and sex included as covariates. The significant taxa identified through this analysis were found to be robust, as they remained significant even after adjusting for these covariates (Supplementary Table 6). Pielou's evenness indices and Shannon index showed lower alpha diversity in children with anaemia or iron deficiency anaemia (Supplementary Table 7). This suggests that bacterial features associated with anaemia in general are strong, while features associated with vitamin A deficiency, stunting, systemic inflammation and parasites exhibited comparatively weaker associations.

**Consumption of multiple micronutrient-fortified rice at school for six months modified children's faecal bacterial composition**
To establish the effect of the intervention strategy on the faecal bacterial composition, we chose first a PCoA representation of Bray-Curtis dissimilarity index of all the children at baseline (*n* = 380) and at the end of the 6-month intervention (*n* = 380) (Fig. 3). The microbial composition changed over time, even in children in the control group who received Placebo rice. Then, to evaluate the difference in the changes in the faecal bacterial diversity in children in each intervention group over time, we used the q-2 longitudinal plugin in to measure alpha and beta-diversity group significance (Table 2). This analysis revealed significant differences not only within each intervention group over time (Shannon index, Pielou's evenness index, Faith's PD distances), but also between the intervention groups (Bray-Curtis, Jaccard, unweighted and weighted Unifrac distances).

Next, we used a linear mixed model to estimate the effect of the intervention on the taxonomic composition of individual faecal microbiota (Fig. 4). The reference used was Placebo rice (*n* = 128). MaAsLin2 analysis identified 42 associations between the UR-original intervention group (*n* = 114) and the genus composition, 22 being negative and 20 positive. Of the 94 associations identified between the

UR-improved intervention group (*n* = 122) and the genus composition, 17 were negative and 77 positive. In all, 136 genera were differentially affected by the treatment. It is interesting to note that most of the bacteria were positively associated with the UR-original intervention (high iron and zinc concentration) compared with the Placebo. Use of the LEfSe representation also identified most of the bacteria as biomarkers of non-anaemic, non-iron anaemic and vitamin A sufficient children (Fig. 2). In contrast, the most frequent bacteria were negatively associated with UR-improved intervention (more diverse micronutrient composition), compared with the Placebo (Fig. 4).

**Predictive functional characteristics of faecal microbiota differed depending on the nutritional status of the children**
Having generated functional profiles of the bacterial communities of each child at baseline and after six months of nutritional intervention, we were interested in establishing a functional hypothesis for the changes observed. We used PICRUSt2 software to predict the functional potential of faecal bacterial communities based on marker gene sequencing profiles[22]. We first compared the differential KEGG pathways between children who differed in their nutritional status, micronutrients, inflammatory status and parasitic infection at the baseline.

The enzymatic pathways were seen to differ depending on the type of intervention as well as on anaemic status, iron deficient anaemia status, vitamin A and zinc status. Children without anaemia had more functions involved in amino acid metabolism than non-anaemic children (Supplementary Fig. 3A). A similar trend was found for iron deficient anaemic children (Supplementary Fig. 3B). Vitamin A deficiency was associated with a reduced abundance of three enzymatic pathways (Supplementary Fig. 3C). Zinc deficiency was mainly associated with increased abundance of the formaldehyde oxidation pathway (Supplementary Fig. 3D).

The difference in the mean proportion was never more than 0.18% for the functions associated with the different metadata cited above. On the contrary, the change in functions over time was stronger and a cut-off effect size > 0.2 was applied to focus on the strongest

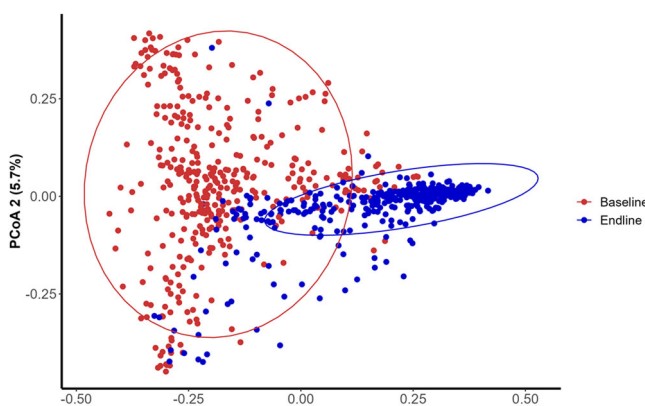

**Fig. 2 | Differential ASVs between the children grouped according to their nutritional status. A** anaemia, (**B**) iron deficiency, (**C**), vitamin A deficiency, (**D**) stunting. The differences were identified using Linear discriminant analysis Effect Size (LEfSe) analysis with an unadjusted *P*-value < 0.05, and a logarithmic LDA (linear discriminant analysis) score of > 2. The cladogram representation illustrating taxonomic levels from the innermost phylum ring to the outermost genera ring. Each circle represents a bacterial member within that level. Circles coloured green or red shows significant enrichment of those taxa in the groups indicated by the legend of each individual cladogram. The number of samples in each subcategory is indicated on each graph. Source data are provided as a Source Data file.

**Fig. 3 | Changes in the structure of faecal microbiota of the 380 children before (red) and after (blue) six months of nutritional intervention.** PCoA representation of Bray-Curtis dissimilarities calculated at the ASVs level. Ellipsoids represent the 95% confidence interval surrounding each sampling time. Significance testing was conducted using PERMANOVA with 999 permutations, and stratification by subject, yielding a *P* value = 0.001. Source data are provided as a Source Data file.

associations. The functions of the Placebo group evolved over time with an increase in the number of functions involved in fatty acid biosynthesis (Supplementary Fig. 3E). At baseline, higher number of functions involved in peptidoglycan biosynthesis and maturation as well as functions related to bacterial metabolism (fermentation, sucrose degradation, *Bifidobacterium* shunt (fructose phosphate pathway) use of acetylene and amino acid synthesis) was observed. No differences between Placebo and UR-improved groups were observed. Only functions involved in fatty acid metabolism were more frequent in the Placebo and UR-original groups at endline (Supplementary Fig. 3F). Stearate biosynthesis was more present in the UR-original group than in the UR-improved group at endline (Supplementary Fig. 3G).

## Discussion
The three objectives of this work were (i) to describe the faecal microbiota of school aged Cambodian children in a large-scale longitudinal cohort, (ii) to investigate associations between age, sex, nutritional status, inflammation and parasitic infection, and faecal microbiota composition and function in this population, and (iii) to explore the impact of a nutritional intervention on changes in faecal microbiota.

**Table 2 | Effect of six months of consumption of multiple-micronutrient fortified rice on alpha (pairwise difference) and beta diversity (pairwise distance) metrics**

| Alpha diversity | | | | |
|---|---|---|---|---|
| | | | **Mann-Whitney U** | **FDR p-value** |
| Shannon index | Intervention group | Placebo vs. UR-improved | 6498 | 0.0672 |
| | | Placebo vs. UR-original | 7844 | 0.2221 |
| | | UR-improved vs. UR-original | 5472 | 0.0104 |
| Pielou's evenness | Intervention group | Placebo vs. UR-improved | 6953 | 0.1349 |
| | | Placebo vs. UR-original | 8544 | 0.0326 |
| | | UR-improved vs. UR-original | 5135 | 0.0016 |
| Faith PD | Intervention group | Placebo vs. UR-improved | 7256 | 0.3346 |
| | | Placebo vs. UR-original | 5923 | 0.0347 |
| | | UR-improved vs. UR-original | 7834 | 0.1400 |
| **Beta diversity** | | | | |
| | | | **Mann-Whitney U** | **FDR p-value** |
| Bray Curtis | Intervention group | Placebo vs. UR-improved | 5871 | 0.0007 |
| | | Placebo vs. UR-original | 9560 | 0.0000 |
| | | UR-improved vs. UR-original | 10471 | 0.0000 |
| Weighted UniFrac | Intervention group | Placebo vs. UR-improved | 7291 | 0.3661 |
| | | Placebo vs. UR-original | 10864 | 0.0000 |
| | | UR-improved vs. UR-original | 10908 | 0.0000 |
| Unweighted UniFrac | Intervention group | Placebo vs. UR-improved | 5928 | 0.0015 |
| | | Placebo vs. UR-original | 8453 | 0.0334 |
| | | UR-improved vs. UR-original | 9636 | 0.0000 |
| Jaccard | Intervention group | Placebo vs. UR-improved | 5768 | 0.0005 |
| | | Placebo vs. UR-original | 8770 | 0.0067 |
| | | UR-improved vs. UR-original | 10055 | 0.0000 |

The q2-longitudinal plugin was used to compare baseline and endline data in terms of alpha and beta diversity group significance using the pairwise difference and pairwise distance tests, respectively. A two-sides Mann-Whitney U test was used. P values were adjusted with Benjamini-Hochberg procedure. Adjusted P values less than 0.05 are considered statistically significant. It evaluated the different changes in faecal bacterial diversity in children in the three intervention groups (Placebo n = 128, UR-improved n = 122, UR-original n = 114) between baseline and endline.

As expected, the faecal microbiota of Cambodian schoolchildren was characterised by a large proportion of Firmicutes, Bacteroidota and Proteobacteria. This is consistent with the result of other studies of the same age groups in Bangladesh and Mexico[23,24]. A study on school-aged children in five Asian countries (China, Indonesia, Japan, Taiwan and Thailand) showed that at family level, the composition of the faecal bacteria varied depending on the cities, but generally comprised a high proportion of *Lachnospiraceae*, *Ruminococcaceae* and to a lesser extent *Prevotellaceae* and *Bifidobacteriaceae*[25]. In our study, the *Lactobacillaceae* family represented a large proportion of the faecal microbiota of Cambodian children, while *Prevotellaceae*, *Ruminococcaceae* and *Lachnospiraceae* represented the third, fourth, and fifth highest proportions after Enterobacteriaceae. In contrast with other Asian countries, the family Bifidobacteriaceae was among the 20 most abundant families, but only in 16th position[25].

The high proportion of *Lactobacillaceae* was unexpected since detection of this family has been uneven in school aged children around the world. For example, no *Lactobacillaceae* were found in children in the Czech Republic, Korea or the USA[26–28] whereas Lactobacillaceae were detected in other studies in China and the USA[29,30]. No trend has been reported in any particular geographical region. A recent study of *Lactobacillus* in the gut and geographical variation based on reanalysis of shotgun sequences, showed that prevalence rates were higher in non-industrialised regions compared to rates in North America[31]. The species belonging to the genus *Lactobacillus* were specific to geographical regions. Indeed, the same authors reported that *Lactobacillus ruminis* was enriched in Asian subjects, while *L. delbrueckii* and *L. casei* were noticeably absent. However,

relative abundance was low, i.e., less than 0.01% of the bacterial population, when the whole study was considered.

In the present study, we found associations between the composition of faecal microbiota and anaemia, but different features emerged, depending on whether we considered anaemia or iron deficiency anaemia. The discrepancies in the bacteria associated with anaemia or iron deficient anaemia are not surprising considering the complex aetiology of anaemia. While it was long considered that anaemia was caused first and foremost by iron deficiency, it is now estimated that less than 50% of anaemic people actually suffer from iron deficiency. In Cambodia, the contribution of iron deficiency to anaemia prevalence is low ( < 10%)[32]. Recently, we reviewed associations between rates of iron deficiency, iron supplementation and gut microbiota reported in the literature[19]. Most studies report strongly diverging results, and neither the form of iron (ferrous sulphate, NaFeEDTA, etc.), nor the experimental model (human, animal, in vitro) could explain these discrepancies. One of the few exceptions was the decrease in *Lactobacillaceae* in the case of iron supplementation, which was observed regardless of whether human, animal or in vitro models were used[19]. In our study, we observed an association between *Limosilactobacillus* and non-anaemia, which is not in line with results in the literature. Indeed, it is generally considered that *Lactobacilli* do not require iron for their growth, and are thus adapted to iron depleted conditions[33].

The regulation of zinc absorption is less complex than the regulation of iron[34]. Zinc deficiency often leads to diarrhoea, and many studies have focussed on the effect of supplementation at different doses in the poultry and pig industries, which have been shown to

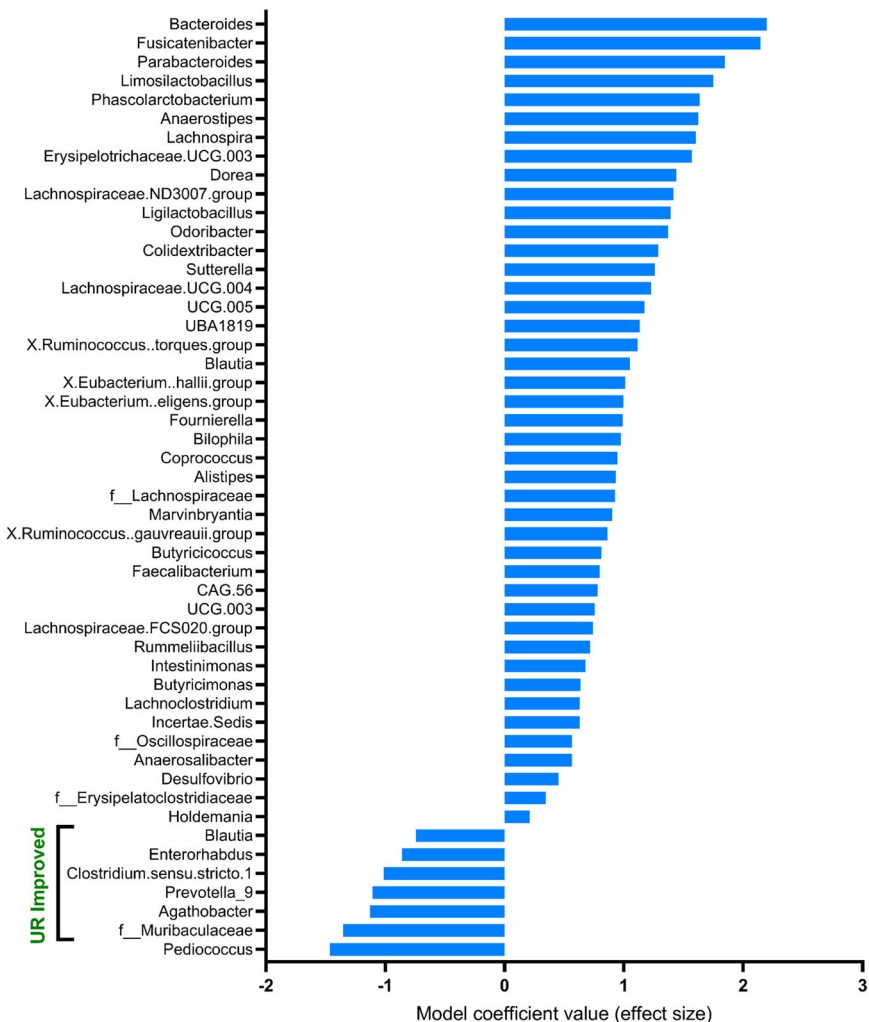

**Fig. 4 | Impact of the nutritional intervention on the taxonomic composition of the faecal microbiota of the 380 children.** Statistical analyses of multivariate associations between UR-original ($n = 114$) and UR-Improved ($n = 122$) intervention groups (compared to the Placebo group, $n = 128$) and gut microbiota composition at the genus level of school-aged children. The analysis was performed using MaAsLin2 with adjustments for multiple comparison ($q$-value < 0.05). Among the 136 genera differentially affected by the intervention, the Top 50 with the largest effect sizes are represented. All detected associations are adjusted for subjects as random effects. Source data are provided as a Source Data file.

cause compositional and functional changes of the gut microbiota[35]. One original study in non-pathogenic conditions of initially germ-free mice inoculated with a mix of different bacteria from the human gut, showed almost no effect of zinc status on bacterial features[36]. This result is in line with the results of the present study, as we observed no within- or between-individual variations in the faecal microbiota of children with different zinc status, pointing to no covariation of microbiota with zinc status. This may be due to the small number of children with normal zinc status in our study. Yet a recent study of 67 schoolchildren in China showed that, despite the absence of beta-diversity difference between control and zinc-deficient children, *Coprobacter*, *Acetivibrio*, *Paraprevotella* and *Clostridium*_XI were more abundant in the zinc deficient group[37].

In our study, a large number of bacterial features including alpha and beta diversity as well as bacterial patterns were associated with vitamin A deficiency, despite the low proportion of vitamin A deficient children (< 8%) at baseline. Normal vitamin A status was associated with Bacilli. Members of *Lactobacillaceae* have been shown to improve vitamin A solubilisation thereby increasing its absorption[38], and have also been shown to convert beta-carotenoids into retinal[39]. We hypothesise that in the case of vitamin A deficiency, bacteria from this order promote the absorption of vitamin A by these two mechanisms, but

further studies are needed to confirm this hypothesis. One limitation of the present study was the use of multiple micronutrient-enriched rice with different micronutrient profiles as well as different amounts of micronutrients, which make it difficult to attribute specific effect to a specific vitamin or mineral.

In the two rice treatments, the type of enrichment was different, UR-original being enriched mainly in iron and zinc, while UR-improved contained additional vitamin A but less iron. Thus, the impact of the two enriched rice formulations on the composition and the predicted functions of the bacterial faecal microbiota were expected to be different. Indeed, the bacterial faecal composition of children who received UR-original was more affected than that of children who received UR-improved. It has been reported in the literature that high iron intakes expose the gut microbiota to iron overload, which has repeatedly been shown to increase enterobacteria, including enteropathogenic *Escherichia coli* and to produce adverse effect including diarrhoea, especially in infants and young children[20]. However, other studies reported no effect of iron consumption on the incidence of diarrhoea[20]. The collection of data on morbidity in the Placebo and intervention groups, specifically diarrheal episodes, would have been a plus.

The association between bacterial abundances and the consumption of UR-original underlines the fact that bacteria belonging to

the family Lachnospiraceae are among the most sensitive to the consumption of micronutrients. The *Lachnospiraceae* family was enriched and associated with baseline absence of vitamin A deficiency, anaemia and iron deficient anaemia. Members of the *Lachnospiraceae* family are known to produce metabolites that are beneficial for the host, but its abundance has been shown to increase in subjects with different diseases, mainly those linked to obesity[40]. In our study, which included no obese children, *Lachnospiraceae* could be a good candidate as a biomarker for absence of vitamin A deficiency and iron deficiency anaemia. To better understand the role of this family in human nutritional status, it would be useful to assess this genus in epidemiological studies characterising the micronutrient status in people with different nutritional statuses.

We expected a change in the functions involved in iron and zinc intake by bacteria with the consumption of the two rice. In addition, we expected a change in vitamin A intake and use between baseline and the end of the intervention We expected a difference in vitamin A intake by bacteria in faecal samples collected from children consuming UR-improved. This was not the case since the differences in predicted functions were surprisingly only related to lipid metabolism. Only a few data are available concerning the bacterial functional aspects of the effect of iron intakes on gut microbiota. For example, it has been shown in an in vitro model of child microbiota that high iron medium reduced not only the butyrate concentration but also the concentration and expression of the gene *butCoAT*, which codes for the butyryl-CoA:acetate CoA-transferase involved in the last step of butyrate production[41]. Different studies using in vitro or animal models associated a modification in the concentration of short chain fatty acids with iron and zinc intakes[42–44]. Another study of the effect of iron supplementation in mice treated with antibiotics showed that iron overload increased the proportion of predicted genes involved in primary bile acid biosynthesis, nitrogen and cyanoamino acid metabolism and pentose phosphate pathways, which was described by the authors as limiting the high oxidative effect of iron[45]. In the present study, we observed no increase in any of these functions. Whole metagenome shotgun sequencing would have been useful to examine functional profiles in depth.

We observed some specific features in the composition of faecal bacteria of children infected with parasites, mainly hookworms, both the *Erysipelotrichaceae* family and Desulfovibrionia were associated with infected children. Intestinal helminths, including hookworms, have been reported to affect faecal bacterial communities[46]. However, in studies involving children, most of the parasites identified belonged to other species, such as *Trichuris*[47]. Additional work on the relationship between hookworms and faecal microbiota is needed to provide a hypothesis on the impact of parasites on the human bacterial compartment and its consequences. Gastrointestinal inflammation has been linked with changes in the structure of the microbiota[48]. In the present study, less than 3% of the children presented a gastrointestinal inflammation, which may explain the lack of association with specific features observed.

One limitation of the study was that no positive control was incorporated in the sequencing.

To sum up, our study was conducted on a cohort of children in one ethno-geographic region where effects of climate, cultures and nationality were consequently not confounding factors. Our results contribute novel information to the description of the faecal microbiota of a cohort of Cambodian schoolchildren, characterised by nutritional, vitamin and mineral status, and the impact of a 6-month nutritional intervention. The faecal microbiota of the Cambodian children was characterised by a surprisingly high proportion of *Lactobacillaceae*. The study revealed associations between faecal microbiota and many micronutrient deficiencies (iron, vitamin A), but not all (zinc deficiency). It also showed that the nutritional intervention did have an impact on the composition of the faecal microbiota.

## Methods

### The FORISCA study
This study was approved by the National Ethic Committee for Health Research (NECHR) of the Ministry of Health, Phnom Penh, Cambodia, the Ministry of Education, Youth and Sports, Phnom Penh, Cambodia, and the Research Ethics Committee of PATH, Seattle, USA. Written informed consent from participants has been obtained. The trial was registered at ClinicalTrials. gov (Identifier NCT0170641 [I https://classic.clinicaltrials.gov/ct2/show/record/NCT01706419?term=NCT01706419&draw=2&rank=1&view=record]). Written informed consent was obtained from their parents or caretakers, and verbal assent was obtained from the participating children prior to enrolment. This present study focusses on a subset of 380 children who participated in the FORISCA project, which was a large, double-blind, cluster randomized, placebo-controlled trial on the impact of fortified rice on the health and cognitive performance of 9500 schoolchildren. The study was conducted in Cambodia in November 2012 and July 2013. A random selection was performed of children for whom all data were available at both points where stool samples had been collected for assessment of parasitic infection and gut inflammation status and who had not taken any antibiotics for the past three months. Exclusion criteria included being less than 6 years old or more than 14 years old, mental or physical disabilities, or severe anaemia (defined as haemoglobin concentration < 70 g/L). There was no participant compensation but children diagnosed with severe anaemia were provided with multiple micronutrient supplements for two months. Six schools in Kampong Speu province that were participating in the United Nations World Food Programme school meal programme were randomly selected for the present trial and placed in three intervention groups named UltraRice Original formulation (UR-original), UltraRice Improved formulation (UR-improved) and Normal rice (Placebo) (Supplementary Table 2)[13].

Anthropometric measurements were taken, and blood and faecal samples were collected at baseline and at the end of the 6-month intervention. Stool samples were collected by providing school children with a sample collection container and requesting them to return the next day to school with a fresh stool sample. Samples were transported in cool boxes containing cool packs (4 °C) to the capital Phnom Penh within 4 h of collection, and further prepared and stored at −20 °C until analysis in Phnom Penh, or transported on dry ice to site of the analysis. After collection of baseline data, all the children received a single 500 mg dose of mebendazole for deworming.

### Anthropometry, blood collection and laboratory analysis
The weight and height of the children were measured without footwear and wearing minimum clothing using standard procedures. Weight was measured once to the nearest 100 g (model 881U scale; Seca, Hamburg, Germany). The accuracy of the scales was checked every day using a set of two calibration weights. Height was measured twice to the nearest 0.1 cm on a wooden stadiometer (UNICEF-Cambodia) and mean values were used. When differences between two measurements of height in the same child exceeded 0.5 cm, the measurements were repeated. Height-for-age Z-score (HAZ) and BMI-for-age Z-score (BAZ) were calculated according to the WHO 2006 reference[49]. Underweight and stunting were defined as BAZ < − 2 and HAZ < − 2, respectively.

Non-fasting blood samples were taken from the antecubital vein in the morning following a standard protocol. Blood (5 mL) was stored in trace-element free vacutainers with no anticoagulant (Vacuette, Greiner Bio One, Austria) at a temperature of < 5 °C. The blood samples were centrifuged (1,300 g for 10 min) and serum samples were aliquoted and stored at −25 °C until analysis. Iron status (ferritin and soluble transferrin receptor), vitamin A status (retinol-binding protein) and biomarkers of inflammation (C-reactive protein (CRP) and α 1-acid-glycoprotein (AGP)) were determined at VitMin laboratory (Willstaett,

Germany). All these proteins were measured using a sandwich enzyme-linked immunosorbent assay (ELISA) technique[50]. Zinc was measured at the National Institute of Nutrition (Hanoi, Vietnam) using a flame atomic absorption spectrophotometer (GBC, Avanta + ) using trace element-free procedures[12]. Faecal calprotectin was measured (Calpro AS, Norway) to estimate the gut inflammation[17].

Faecal parasites were analysed as previously described using the Kato-Katz technique to determine helminth infection by the National Center for Parasitology, Entomology and Malaria control (CNM), Phnom Penh, Cambodia, and recorded as the number of eggs per gram of faeces[16].

Logistic mixed models were performed to assess the impact of the intervention on child growth and micronutrient status using R 4.1.0. Fixed effects were the intervention group, the time of measurement, their interaction, the age group and the sex. Random terms for child ID and schools were used in the models. The interaction "time of measurement × group of intervention" estimated the impact of the intervention on child growth and micronutrient status. For mixed models, results are expressed as odds ratios (OR) for the interaction "time of measurement × group of intervention" with 95% confidence intervals (CI) and corresponding p-values.

## Taxonomic profiling of the gut microbiota

Faecal samples for taxonomic profiling of the gut microbiota were collected from the three groups at baseline and endline. Faecal samples were stored at −80 °C until DNA extraction. DNA was extracted using the QIAamp DNA Stool Mini kit (ID: 51604, Qiagen, Les Ulis, France) according to the manufacturer's protocol[51]. The V3-V5 hypervariable region of bacterial 16SrRNA was sequenced in paired-end mode (2×300 bp) on the MiSeq platform (Illumina, performed by the Research and Testing Laboratory in Lubbock, Texas, US) using primers 357 F (5′-CCTACGGGAGGCAGCAG-3′) and 926 R (5′-CCGTCAATTCMTTTRAGT-3′)[52]. Different intervention groups as well as time points (baseline and endline) were randomized, pooled in four batches of 190 samples and analysed into four runs to ensure that batches did not contain either all baseline or all intervention group samples. A no template control was included at the PCR step and sequenced along with the samples. The 16 S rRNA gene sequencing data generated in this study have been deposited in the National Centre for Biotechnology Information database under accession number PRJNA882252. Demultiplexed sequences were obtained and the Divisive Amplicon Denoising Algorithm2 (DADA2) R package was used to remove low quality reads, to de-noise and remove chimeric sequences, and the data were then sorted into unique amplicon sequence variants (ASVs)[53]. Due to the small overlap between forward and reverse reads following DADA2 trimming and quality filtering, only forward reads were used for analysis. The outputs from DADA2 (ASVs) were assigned to taxonomies using the Ribosomal Database Project (RDP) Naïve Bayesian classifier[54] using the DADA2 formatted training sets for SILVA 16 S rRNA database version 138.1 (https://benjjneb.github.io/dada2/training.html)[53–55]. Chloroplast and mitochondria sequences were removed. Sequences were rarefied at 3500 reads per sample to determine bacterial alpha and beta-diversity using QIIME2 (2021.11)[56]. The core microbiota, represented by taxa occurring in at least 95% of the subjects who participated in the study, was also assessed at the genus level using the microbiome R package[57,58]. Differences in alpha diversity were statistically assessed using the Kruskal-Wallis test, with significance set at $P < 0.05$. Additionally, a linear mixed model incorporating covariates such as age and sex was used to further validate the robustness of significant features. Significance was set at $P < 0.05$. Beta-diversity group significance was assessed using permutational analysis of variance (PERMANOVA) with 999 permutations. Bray-Curtis at the baseline and endline was also compared with PERMANOVA with 999 permutations while accounting for the stratification by subject using adonis2 function from the vegan R package[59].

Both tests were used on ASVs levels at the baseline for all variables and at endline to explore the effect of intervention on alpha and beta diversity.

Additionally, ASVs with a prevalence of < 5% were removed from taxonomic analyses and linear discriminant analysis effect size (LEfSe) was applied (Wilcoxon $P$-value < 0.05, logarithmic LDA (linear discriminant analysis) score > 2) to identify the biomarkers in different groups of metadata[60]. Following the LEfSe analysis, taxa showing significant differences were further tested in a linear mixed model with log-transformed values adjusted for covariates such as age and sex. P-values were corrected for multiple comparison using the Benjamini-Hochberg procedure ($q$-value < 0.1). The effect of multiple-micronutrient fortified rice (UR-original and improved) on gut microbiota composition was investigated using the q2-longitudinal plugin to compare baseline and endline data in terms of alpha and beta diversity group significance using pairwise difference and pairwise distance tests, respectively[61]. We conducted a multivariate association analysis to assess the impact of the intervention on microbiome composition at the genus level. In this analysis, a linear mixed model effect analysis was performed using the MaAsLin2 R package[62]. Only bacterial genera present in at least 10% of the participants in the study were included in the analysis. Subsequently, the bacterial genera underwent total sum scale (TSS) transformation without normalization for the analysis. The intervention group and time (baseline and endline) were used as a fixed effect while subjects were used as a random effect. A q-value threshold of 0.25 was applied to determine statistical significance. Data supporting the findings of this study and script used for data analysis are accessible at https://github.com/yoh-s/FORISCA.git.

## Functional potential of the 16 S rRNA-predicted taxa identified in the samples

Phylogenetic Investigation of Communities by Reconstruction of Unobserved States 2 (PICRUSt2) software was used with default parameters to predict the functional potential of the 16 S rRNA[22]. In PICRUSt2, sequence variants are placed in a phylogenetic tree using Massively Parallel Evolutionary Placement of Genetic Sequences (EPA-NG)[63] and processing, analysing and visualising phylogenetic (placement) data (gappa)[64]. Genomic hidden states were predicted using efficient comparative phylogenetics on large trees (castor)[65], and MetaCyc pathway abundances[66] were inferred using MinPath[67].

Statistical analysis of metagenomic profiles (STAMP v2.1.3) was used to test group significance between-groups. Welch's t-test was performed with Benjamini-Hochberg (BH) False Discovery Rate correction. Adjusted p-value (BH) less than 0.05 were considered statistically significant.

## Statistics & reproducibility

The sample size of FORISCA study was calculated based on the intention to detect changes in anaemia prevalence, haemoglobin concentration and a number of micronutrient statuses. The sample size of 500 children per intervention group was decided on the ability to detect a difference in haemoglobin concentration between placebo and intervention groups of at least 3 g/L, assuming a power of 0.90, a statistical significance of 0.05 and a SD of 20 g/L. Other outcomes all needed smaller sample sizes[12]. The randomization, blinding and subsample general information are described in the paragraph "The FORISCA study". No data were excluded but some data are not available for all children due to obtaining not enough sample. The number of schoolchildren included for each analyse is presented in the figures.

Statistical analyses of the effect of the intervention on child growth and micronutrient status is described in details in the paragraph "Anthropometry, blood collection and laboratory analysis". Statistical analysis for the link between faecal microbiota, nutritional, micronutrient, inflammatory and anthropometrical status as well as the effect of intervention on the composition of the faecal microbiota

are detailed in the paragraph "Taxonomic profiling of the gut microbiota". Statistics related to the evaluation of the functional potential of the faecal microbiota and its relation with micronutrient status and the effect of the intervention are described in the paragraph "Functional potential of the 16 S rRNA-predicted taxa identified in the samples".

## Reporting summary

Further information on research design is available in the Nature Portfolio Reporting Summary linked to this article.

## Data availability

The 16SrRNA genes sequencing data generated in this study have been deposited in the National Centre for Biotechnology Information database under the accession number PRJNA882252. The healthy volunteer-related clinical trial raw data and the processed data are provided with this paper in the Source Data file. Source data are provided with this paper.

## Code availability

All codes have been deposited on GitHub repository https://github.com/yoh-s/FORISCA.

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

## Acknowledgements

We would like to thank all the schoolchildren and their parents in Kampong Speu province for participating in the study. We are also very grateful to school teachers, school monitors, and to the staff of DFPTQ, IRD, WFP, PATH, National center for malaria (CNM), Royal University of Phnom Penh (RUPP), and the Kampong Speu Health and Education Department for their support in the field and data collection.

Funding came from the United States Department of Agriculture/ FAS through a grant (FFE-442-2012/03800, 10.608) to PATH and internal funding came from WFP (through the WFP/DSM partnership), and from the French National Research Institute for Sustainable Development (IRD).

## Author contributions

Conceptualization, J.B., K.B., C.C., F.W. and C.H. Methodology, Y.S., V.G., R.M., V.C. Formal Analysis, Y.S., V.G., D.K.D.K, W.T., V.C., S.F. Investigation, K.K., M.P., M.F. Resources, K.K., M.P., M.F. Data Curation, Y.S., V.G., D.K.D.K. Writing – Original Draft, Y.S., V.G. Writing – Review & Editing, W.T., S.F, J.B., M.C.P., F.W., and C.H. Supervision, M.C.P., C.C., F.W., and C.H. Funding Acquisition, C.C., F.W.

## Competing interests

The authors have no conflicts of interest to declare.

## Additional information

¹Qualisud, Univ Montpellier, Avignon Université, CIRAD, Institut Agro, IRD, Université de La Réunion, Montpellier, France. ²Department of Fisheries Post-Harvest Technologies and Quality Control, Ministry of Agriculture, Forestry and Fisheries, Phnom Penh, Cambodia. ³United Nations World Food Programme, Phnom Penh, Cambodia. ⁴Section Infectious Diseases, department of Health Sciences, Faculty of Earth and Life Sciences, VU University Amsterdam, Amsterdam, The Netherlands. ⁵Present address: Zane Cohen Centre for Digestive Diseases, Mount Sinai Hospital, Toronto, ON, Canada. ⁶Present address: Division of Gastroenterology, Temerty Faculty of Medicine, University of Toronto, Toronto, ON, Canada. ⁷Present address: MoISA, Univ Montpellier, CIHEAM-IAMM, CIRAD, INRAE, Institut Agro, IRD, Montpellier, France. ⁸Present address: SESSTIM, INSERM, IRD, Aix Marseille Univ, Marseille, France. ⁹These authors contributed equally: Frank T. Wieringa, Christèle Humblot. ✉e-mail: Christele.Humblot@ird.fr

