## [Peer Review File · Nature Communications]

Faecal microbiota of schoolchildren is associated with nutritional status and markers of inflammation: a double-blinded cluster-randomized controlled trial using multi-micronutrient fortified rice.REVIEWER COMMENTS

Reviewer #1 (Remarks to the Author):

Seyoum et al propose to investigate interactions between the microbiota and various demographic and nutritional characteristics in school-age children selected as a substudy population from a larger trial of fortified foodstuffs in Cambodia. Notably, the fortifications examined in the FORISCA study are largely micronutrient-related, thus Seyoum et al highlight the worthy goal of understanding interactions specifically between the microbiota and micronutrient deficiency status. Overall, I feel the intent of the work described in Seyoum et al is absolutely important to the field; however, the level of sophistication and descriptive nature of the work, plus its presentation and interpretation do not meet the standards for publication in Nature Communications.

General Comments:

1. Introduction – the authors claim that information on “bacterial microbiota in with acute malnutrition are scarce.” Although I agree this area is understudied, there are significant published studies in the literature in India, Bangladesh, Malawi, and elsewhere from Jeff Gordon’s Lab at Washington University in St. Louis (see Smith et al, 2013; Subramanian et al, 2014; Chen et al, 2021, for example), Ramnik Xavier’s Lab at the Broad (Vatanen et al, 2022), and others. I feel some of these studies should be acknowledged.
2. Introduction – numerous studies link the gut microbiota with micronutrient deficiencies in the early days of gnotobiotic research. Many resources are reviewed in Smith, McCoy, and MacPherson, 2007.
3. FORISCA study- it’s unclear how this substudy fits into the broader context of the study. The authors assume the reader understands details of the language, design, and length of the broader study, but at least a brief introduction is warranted here. For example, apparently two treatments were applied in the FORISCA study, but the authors frequently switch between discussing them as one and as two separate treatments in the text and tables without details on their differences.
4. The analytical approach in the broader sense (through qiime) and specifically in terms of the analyses selected, are extremely descriptive and routine. Without a detailed justification of the reasoning for each analysis and the question it addresses, relevant to the questions posed, the paper reads as a very routine, descriptive analysis straight from a qiime tutorial. The figures also suffer from this same deficiency. A search of the Nature Communications manuscript catalog for “microbiota amplicon” returns a high-quality set of papers with similar underlying data but with well-explained and reasoned (and presented) analyses.

Specific Comments:

1. Line 106: "Supplementary Table2" is referenced – where is Table1?
2. Line 119: What type of sequencing reads were generated? Using what strategy for library preparation. What instrument? Etc?
3. Line 127: "Data not show" for a positive result is not acceptable.
4. Line 131-133: The authors reference PERMANOVA but then Kruskal-Wallis in the parenthetical? There's detail here missing – the application of these two statistical techniques are not clearly described.
5. Line 140 (and others): numbers of samples tested in subcategory analyses are never explained. What's the "n" for these comparisons?
6. Line 158: Figure2 has labels and components that are partially cropped or otherwise obscured.
7. Lines 191 – 206, specifically 191-192: The authors described the confounding nature of changes in the microbiota in the placebo group. This is not surprising, and using LMEs as a strategy to address this is reasonable. However, the authors insufficiently describe their strategy to fairly evaluate whether they have properly controlled for the "control" group.
8. Line 200: the authors suggest they're using LMEs, then refer to correlations. Are they correlations or coefficients?
9. Line 209: PICRUST was employed, but why? The authors don't justify this technique or why it was employed.
10. Line 223: "gastrointestinal inflammation, which has never been significantly associated with faecal bacterial composition in all previous analyses..." These types of statements are too definitive and in this case, untrue. See the literature from Andreas Baumler's lab on inflammatory pathogens and microbiota configuration, plus a broad immunological literature.
11. Lines 273-275: The authors refer to the work in gnotobiotic animals describing micronutrient-microbiota interactions as not pertinent. I agree that more human studies relating micronutrient status to the microbiota are necessary, but there are ethical considerations that make the ideas discussed in the animal study literature very important.
12. Line 314: "first to describe" – this does not constitute impact.

Reviewer #2 (Remarks to the Author):

Summary

This study is a secondary analysis of data and specimens originating from the FORISCA trial (2012-13), a double-blinded, cluster-randomized, placebo-controlled trial including 9500 children that tested the impact of rice fortified with vitamins and minerals on nutritional status and anthropometry of children aged 6–14 years in Cambodia. The trial found nutritional improvements, but a perplexing increase in hookworm infections and increased systemic and intestinal inflammation due to the intervention. This secondary analysis sought to understand the microbiota-mediated mechanisms for the FORISCA trial results. The authors examined microbiota composition in 380 children enrolled in FORISCA at baseline and again at the end (endline) of the 6-month fortified rice intervention. Associations between microbiota composition and inferred (PICRUST) microbiota function, based on 16S rRNA sequencing of 760 fecal samples, and the following factors: age, sex, nutritional status (underweight, stunting), micronutrient deficiencies, inflammation (systemic, gut), and parasitic infection, were evaluated. The microbiota was taxonomically characterized at the genus or higher. Child fecal microbiota exhibited a higher than expected abundance of Lactobacillaceae. There were associations between the microbiota and micronutrient deficiencies and the fortified rice trial intervention.

Major comments

P2 There are a number of papers that investigate the relationship between the microbiota and malnutrition/undernutrition in children from LMICs, especially those papers produced by the Jeff Gordon and Indi Trehan groups at Washington University and AfriBiota, among others.

P5 The prevalence of iron and zinc deficiencies were 51% and 89%, in the sample of 380 children, but the population-wide zinc deficiency prevalence estimate of 50% was much lower compared to the sample. Vitamin A deficiency was also lower. Can the authors explain the significant differences in micronutrient deficiencies between the population and their sample? Was this likely to be random or systematic sampling error in the selection of subjects from the larger trial into this study?

P5 Suggest moving the short paragraph (“The impact of the intervention ...”) describing the impact of the intervention on micronutrient deficiencies to the introduction, and add results for stunting, which seem to be missing from the FORISCA trial summary in the introduction.

P5 Was there other information about the included children, for example, information on antibiotic or other medication use, non-rice related dietary factors, or underlying illnesses (e.g., diarrhea). As the authors mention in the introduction, these could be important confounders in these observational data stemming from a clinical trial. It would be very useful to present these data, indicate whether these factors were included in the analysis, or add to a limitations section in the discussion.

P5 The authors provide an estimate of the average read counts per sample, but there are no details about the range or distribution of read counts (some datasets could be very small or large), the

quality of the reads (e.g., how many reads had to be quality filtered), and there is no indication of results for negative/process or positive controls.

P5 What was the rationale for a 95% threshold for shared microbiota? This is a very high criterion to use. The endline microbiota results for the core microbiota were not shown, was there a reason for this? Did the authors consider whether the “core genera” were the same in each intervention group, especially for the endline sample? It seems the endline core microbiota members could differ due to the intervention received, and this could be important in the context of this report. Alternatively, the core changes could be due to age, and be the same across the trial intervention arms.

P6 Could the authors provide a rationale for why so many beta diversity metrics were tested? If the authors believe the Bray-Curtis or Jaccard is a better fit for their data then it would be helpful to explain why and how this helps their interpretation. Otherwise, this may increase, unnecessarily, the possibility of a false positive.

P6 It seems that age is related to differences in microbiota diversity. Since, this is the case, then age should be accounted for when evaluating changes in microbiota diversity related to the different micronutrient deficiencies. If the authors did account for this, then please clarify.

P6 “Pielou’s evenness was significantly different in children with or without anaemia, iron deficiency anaemia, vitamin A deficiency.” The authors state that these values of microbiota alpha diversity are different, but would be informative to state how they differ (e.g., was the alpha diversity higher or lower in younger versus older children). Multiple associations (e.g., diversity measures and micronutrient deficiency results) are described in the manuscript, but often the direction of the association is omitted.

P6 Was PERMANOVA used for the alpha diversity comparisons, it is not clear from the first sentence on the page. It is clear PERMANOVA is used for testing differences in beta diversity in the methods, but that sentence is confusing.

P6 Add reference to Figure 3 -- related to differences in beta diversity by age. The PCoA plot for the age groups is very different. Were the baseline and endline samples run separately, in batches, or all mixed together? Is it possible that some of the divergence is due to differences in run effects?

P7 The authors have already established that there are differences in microbiota composition by age and sex. Lefse does not allow for the inclusion of confounding variables such as age and sex, consider reporting the Maaslin results to account for these factors.

P8 Based on type of enrichment made to the rice, did the authors have an a priori idea of what functional pathways would be impacted by the enriched rice formulations? The interpretation of links between microbiota functions and fortified rice could also be strengthened in the discussion.

P8 For the PICRUSt analysis, and supplemental results, consider restricting the results by significance or highest effect size. This will shorten the Supplemental Figure 3 (especially A and G) and make it readable. Was PICRUSt only performed on the 16S rRNA data from the endline sample? Would it be helpful to compare the functions that are found in both time-points to separate out common functions that persist over time, and then focus on those that differ, especially in the comparison across intervention groups? This might reduce the number of variables to focus on.

P12 Could the authors comment on whether the taxa or functions associated with inflammation and parasitic infection overlap?

P13 The authors should add a section in the methods on how the stool specimens were collected. They should also comment on whether the samples were stored stably and if they were subject to any freeze-thaw cycles.

P13. Please provide a reference(s) for the weight and height “standard” measurement procedures.

P14. The logistic mixed models methods and analysis section refers to Maaslin, please make that clear in the methods. Provide a cut-off for interpreting coefficients, such as effect size. Again, this will cut down on the variables for interpretation.

P14 Add a section on limitations to the discussion.

P15 “functional potential of the 16S rRNA-predicted taxa identified in the samples.”

Figure 2B. Not clear what yes/no refers to in the figure, please add iron deficiency to the graph labels directly. Same for Supplemental Figure 2A.

Table 1. The legend should be clarified; PERMANOVA is used to test differences in beta diversity, but not alpha diversity.

Table 1. Shannon index and Evenness values should be presented for each group so the reader can see the magnitude of the differences in the values

Minor comments

The first sentence of the introduction can be removed.

P1 repetitively -> repeatedly

P1 authors -> investigators

P4 “39,355 read counts per sample”

P8 Revise linear to logistic mixed effect model.

P12 revise confusing -> confounding

There is a figure 5 legend shown on page 35, but no figure.

Reviewer #3 (Remarks to the Author):

Title: Faecal microbiota of schoolchildren is associated with nutritional status, micronutrient status and markers of inflammation. (NCOMMS-22-45889-T)

Summary: This paper sought to describe the gut microbiota of Cambodian school children and investigate associations with age, sex, nutritional status, inflammation, and parasitic infection.

Reviewer's Comments:

Validity:

1. This manuscript did not have any major flaws prohibiting its publication.

Originality & significance:

1. The results presented are of interest to researchers in nutrition and gut microbiome in children.

2. Cambodia is an understudied setting for gut microbiome research, making this study a valuable addition to the literature.

Data & methodology:

1. What iron formulation was used to fortify the rice?

2. Lines 322: What was the inclusion and exclusion criteria for the parent trial? Were there any additional inclusion or exclusion criteria for being included in the substudy? Was antibiotic use taken into account?

3. Lines 338-341: What equipment were used for measuring weight and height?

4. Lines 333-335: Please give more details on the fecal sample collection. What kit was or method was used? Were participants trained on sample collection?

5. Line 345-348: What methods were used for the nutritional assays? Only the assay for zinc is described.

6. Line 208: Please reconsider the use of PICRUSt, as it was not validated for children and may not be appropriate.

7. Lines 364-380: Were chloroplast and mitochondrial sequences filtered out? Please add.

8. Were there any adverse effects? Particularly in the iron groups? (See Clarity & Context below)

Appropriate use of statistics and treatment of uncertainties:

1. No comments

Suggested improvements:

1. Introduction: Suggest shortening this with the following outline: Paragraph I: Defining the gut microbiota and its roles; II: What affects the gut microbiota and what is currently known about

undernutrition and the gut microbiota in children (see reviews by Million et al. 2017, Barratt et al. 2022, Robertson 2020); III: Parent trial (FORISCA) details; IV. The gap and the objective of your study.

2. Results:

a. Lines 105-111: Please add the full age range of the children.

b. Lines 113-116: “UR-original” and “UR-improved” are not previously defined (perhaps due to the methods section being at the end). Please check and introduce/describe at first mention in the results section. Also, a placebo is mentioned later, and it is unclear if the placebo is one of the above interventions.

c. Line 122: I believe the term “Bacteroidetes”, which is synonymous with Bacteroidota, is more commonly used; please consider using this term instead.

References:

1. See above

Clarity and context:

1. Considering the intervention of interest is a high-iron / fortified rice, it would be very interesting to understand how your results compare to the results described by the Zimmermann group (Jaeggi et al. 2015, Paganini 2017 and 2019, etc.) and to put this study in the context of iron interventions and the gut microbiome.

Point by point response to the reviewers:

NCOMMS-22-45889A: Faecal microbiota of schoolchildren is associated with nutritional status, micronutrient status and markers of inflammation.

We would like to thank the reviewers for their interesting comments on the manuscript. We have taken into account all the reviewer's remarks. Please find attached our detailed answer to the reviewers' comments.

We hope that these revisions will be suitable to you.

REVIEWER COMMENTS

Reviewer #1 (Remarks to the Author):

Seyoum et al propose to investigate interactions between the microbiota and various demographic and nutritional characteristics in school-age children selected as a substudy population from a larger trial of fortified foodstuffs in Cambodia. Notably, the fortifications examined in the FORISCA study are largely micronutrient-related, thus Seyoum et al highlight the worthy goal of understanding interactions specifically between the microbiota and micronutrient deficiency status. Overall, I feel the intent of the work described in Seyoum et al is absolutely important to the field; however, the level of sophistication and descriptive nature of the work, plus its presentation and interpretation do not meet the standards for publication in Nature Communications.

We thank the reviewer for the useful comments we addressed in the best of our possibilities, which improves substantially the manuscript. We hope the update manuscript meets now the standards for publication in Nature Communications.

General Comments:

1. Introduction – the authors claim that information on “bacterial microbiota in with acute malnutrition are scarce.” Although I agree this area is understudied, there are significant published studies in the literature in India, Bangladesh, Malawi, and elsewhere from Jeff Gordon's Lab at Washington University in St. Louis (see Smith et al, 2013; Subramanian et al, 2014; Chen et al, 2021, for example), Ramnik Xavier's Lab at the Broad (Vatanen et al, 2022), and others. I feel some of these studies should be acknowledged.

Thank you for your feedback, we modified the introduction following the suggestions. The text reads now:

P3 L70-

While a large body of literature reports associations between the human microbiome and obesity, even if the results are inconsistent [5], data on bacterial microbiota in children with acute malnutrition are relatively scarce but once again underline the importance of gut microbiota [6]. Indeed, severe and acute malnutrition is associated with significant relative microbiota immaturity [7,8]. The use of gnotobiotic mice also identified the gut microbiome as a causal factor in severe acute malnutrition, in addition to insufficient nutrient intake [9]. Significantly, the use of microbiota-targeted complementary food was more efficient than ready-to-use supplementary foods in increasing weight in moderately malnourished Bangladeshi children [10].

2. Introduction – numerous studies link the gut microbiota with micronutrient deficiencies in the early days of gnotobiotic research. Many resources are reviewed in Smith, McCoy, and MacPherson, 2007.

Thank you for your remark, the paragraph was modified following your suggestions. The text reads now:

P4 L95-

Numerous studies have investigated the role of gut microbiota in micronutrient status. Indeed, early work comparing germ-free and conventional animals showed a small but crucial role of the gut microbiota in the availability of different B-vitamins (B5, B8, B9, B12 but not B1) in animals fed a deficient diet [17]. Zinc requirements were also found to be reduced in the germ-free state, which was not the case for iron requirements [17]. Some evidence for associations between micronutrient status and specific gut microbiota composition has been reported, for example a decrease in the relative abundance of Bifidobacterium has been associated with iron supplementation [18,19].

3. FORISCA study- it's unclear how this substudy fits into the broader context of the study. The authors assume the reader understands details of the language, design, and length of the broader study, but at least a brief introduction is warranted here. For example, apparently two treatments were applied in the FORISCA study, but the authors frequently switch between discussing them as one and as two separate treatments in the text and tables without details on their differences.

We agree with the reviewer that the FORISCA trial was described in too few details in the introduction and results sections, since the material and methods is at the end. We've added details on the FORISCA trial in both the introduction and results sections. The text is now:

P 4 L 85-

The aim of the 'Fortified Rice for School Children in Cambodia' (FORISCA) project was to assess the impact of rice fortified with different levels of vitamins and minerals on nutritional status, development and anthropometry of schoolchildren aged 6–14 years, who received fortified rice or normal rice for six months. The FORISCA trial showed that daily consumption of three types of fortified rice with different micronutrient content provided through a school meal programme improved the status of most micronutrients [12,13], had a small but nevertheless significant impact on cognitive development [14], but simultaneously increased the prevalence of hookworm infection [15]. Evidence for gut and systemic inflammation was found in respectively 5.4% and 39.5% of the children [16]. The three different formulations of fortified rice had different impacts on these parameters [13,15].

P5 L123-

This study involved a subset of 380 children randomly selected from the parent FORISCA double-blind cluster randomized controlled trial using multi-micronutrient fortified rice (ClinicalTrials.gov identifier NCT01706419). The characteristics of the participants at baseline are listed in Supplementary Table 1.

P6 L133-

The 380 schoolchildren had been randomized (with school as cluster) to receive either normal rice (Placebo), or one of the two types of fortified rice: UltraRice Original formulation (UR-original) or UltraRice Improved formulation (UR-improved) for a period of six months. UR-original was fortified with four micronutrients (iron, zinc, vitamin B1 and B9). UR-improved

had four additional vitamins (vitamin A, vitamins B3, B6 and B12). Their final composition differed slightly, with UR-original providing slightly more iron and zinc than UR-improved (Supplementary Table 2).

P6 L 141-

The impact of the two interventions on micronutrient status differed, for details, see supplementary Table 3.

P 10 L 254-

MaAsLin2 analysis identified 42 associations between the UR-original intervention group (n = 114) and the genus composition, 22 being negative and 20 positive. Of the 94 associations identified between the UR-improved intervention group (n = 122) and the genus composition, 17 were negative and 77 positive. In all, 265 genera were differentially affected by the treatment. It is interesting to note that most of the bacteria were positively associated with the UR-original intervention (high iron and zinc concentration) compared with the Placebo. Use of the LEfSe representation also identified most of the bacteria as biomarkers of non-anaemic, non-iron anaemic and vitamin A sufficient children (Figure 2). In contrast, the most frequent bacteria were negatively associated with UR-improved intervention (more diverse micronutrient composition), compared with the Placebo (Figure 4).

4. The analytical approach in the broader sense (through qiime) and specifically in terms of the analyses selected, are extremely descriptive and routine. Without a detailed justification of the reasoning for each analysis and the question it addresses, relevant to the questions posed, the paper reads as a very routine, descriptive analysis straight from a qiime tutorial. The figures also suffer from this same deficiency. A search of the Nature Communications manuscript catalog for “microbiota amplicon” returns a high-quality set of papers with similar underlying data but with well-explained and reasoned (and presented) analyses.

We thank the reviewers for this comment and we rewrote the results section, modifying the titles and adding justification of the reasoning for each analysis and the question it addresses. We also modified the figure captions to be more explicit.

In addition, the reviewer 2 underlined that the differences in microbiota composition by age and sex should be included as confounding variables. For that reason, we ran linear mixed model on taxa identified by LefSE analysis as differentially abundant depending on the nutritional, micronutrients, inflammatory and parasitic infection statuses. We added two supplementary tables (Supplementary table 4, 5) and one paragraph as follow:

P9 L229-

Age and sex were identified as potential confounding variables, as they were associated with differences in beta-diversity (Table 1) and in composition (Supplementary Figure 2). Therefore, the significant features identified by LefSE analysis were integrated in a linear mixed model, with age groups (6-9 and 10-14 years old) and sex included as covariates. The significant taxa identified through this analysis were found to be robust, as they remained significant even after adjusting for these covariates (Supplementary Table 4). Pielou's evenness indices and Shannon index showed lower alpha diversity in children with anaemia or iron deficiency anaemia (Supplementary Table 5). This suggests that bacterial features associated with anaemia in general are strong, while features associated with vitamin A deficiency, stunting, systemic inflammation and parasites exhibited comparatively weaker associations.

Specific Comments:

1. Line 106: “Supplementary Table2” is referenced – where is Table1?

All tables numbers were double checked and corrected when required.

2. Line 119: What type of sequencing reads were generated? Using what strategy for library preparation. What instrument? Etc?

We carefully checked other articles from Nature Communications, which confirmed that the details (library preparation, instrument, etc...) are in the material and methods at the end of the manuscript. For more clarity we added a sentence to precise the type of reads generated in the results section as follow:

P6 L148-

To investigate the faecal bacterial composition of the 380 schoolchildren, we applied 16S rRNA amplicon sequencing at baseline and again after six months of nutritional intervention.

All details are in P19 L527- in the material and methods section.

3. Line 127: “Data not show” for a positive result is not acceptable.

Thank you for your comment. We represented the core microbiota at the endline as a whole, which is not relevant. We wanted to underline the evolution of the microbiota along time, which is done a better way in the figure 3. As a deep investigation of the effect of the intervention on the faecal microbiota is described in the paragraph entitled “Consumption of multiple micronutrient-fortified rice at school for six months modifies the children’s faecal bacterial composition.”, we considered removing this sentence, and keep the detailed analyses of the effect of the intervention, which is presented **P9 L240-**

4. Line 131-133: The authors reference PERMANOVA but then Kruskal-Wallis in the parenthetical? There’s detail here missing – the application of these two statistical techniques are not clearly described.

Thank you for the comment. Details were only in the material and method section, we modified the text and the figure legend to be clearer about the application of the two different statistical techniques as follow:

P7 L 162-

Differences in microbiota among children grouped according to age, sex, nutritional, micronutrients, inflammatory status and parasitic infection were evaluated based on their alpha-diversity using Kruskal Wallis statistical analysis and on their beta-diversity using Permutational Multivariate Analysis of Variance (PERMANOVA) analysis (Table1).

P30

Table 1: Baseline alpha- and beta-diversity (pairwise distance) metrics of the 380 children grouped according to age, sex, nutritional micronutrient and inflammatory status and parasitic infection.

Differences in alpha-diversity were estimated using a one-way Kruskal Wallis test. The direction of the change is indicated by “<”. Differences in beta-diversity were estimated using PERMANOVA one-way analysis with 999 permutations. The number of samples in each subcategory is indicated in brackets. For all alpha diversity measures, the median \pm standard deviation is presented alongside the number of participants (n, median \pm standard deviation).

5. Line 140 (and others): numbers of samples tested in subcategory analyses are never explained. What’s the “n” for these comparisons?

Thank you for your remark. The number of samples has been added in all the tables and figures of the manuscript and in the corresponding text.

6. Line 158: Figure2 has labels and components that are partially cropped or otherwise obscured.

Sorry for the figures, but in our version the quality is high (600 dpi). Maybe it’s due to the pdf construction during the submission process.

7. Lines 191 – 206, specifically 191-192: The authors described the confounding nature of changes in the microbiota in the placebo group. This is not surprising, and using LMEs as a strategy to address this is reasonable. However, the authors insufficiently describe their strategy to fairly evaluate whether they have properly controlled for the “control” group.

Thank you for the remark. The control group was the group of children receiving normal rice, not fortified with vitamins and minerals. In all other aspects, this group of children was identical to the groups of children receiving the fortified rice. We have tried to make this more clear in the manuscript and included it in the first result paragraph as follow:

P5 L123-

This study involved a subset of 380 children randomly selected from the parent FORISCA double-blind cluster randomized controlled trial using multi-micronutrient fortified rice (ClinicalTrials.gov identifier NCT01706419). The characteristics of the participants at baseline are listed in Supplementary Table 1.

P6 L 133-

The 380 schoolchildren had been randomized (with school as cluster) to receive either normal rice (Placebo), or one of the two types of fortified rice: UltraRice Original formulation (UR-original) or UltraRice Improved formulation (UR-improved) for a period of six months. UR-original was fortified with four micronutrients (iron, zinc, vitamin B1 and B9). UR-improved had four additional vitamins (vitamin A, vitamins B3, B6 and B12). Their final composition differed slightly, with UR-original providing slightly more iron and zinc than UR-improved (Supplementary Table 2).

8. Line 200: the authors suggest they’re using LMEs, then refer to correlations. Are they correlations or coefficients?

Certainly, we employed a linear mixed model to assess the impact of the intervention on the taxonomic composition of individual faecal microbiota using MaAsLin2. In this statistical analysis, the coefficient values signify the strengths of association (either positive or negative)

between the phenotype and the covariate. To prevent confusion, we have substituted the term "correlation" with "association" throughout the text.

The text reads now:

P10 L252-

Next, we used a linear mixed model to estimate the effect of the intervention on the taxonomic composition of individual faecal microbiota (Figure 4). The reference used was Placebo rice (n = 128). MaAsLin2 analysis identified 42 associations between the UR-original intervention group (n = 114) and the genus composition, 22 being negative and 20 positive. Of the 94 associations identified between the UR-improved intervention group (n = 122) and the genus composition, 17 were negative and 77 positive.

9. Line 209: PICRUST was employed, but why? The authors don't justify this technique or why it was employed.

Thank you for the remark. The PICRUST was employed to establish functional hypothesis on the changes observed at compositional level between schoolchildren with different nutritional status and between schoolchildren for the different intervention groups. This could provide insights into the potential metabolic pathways and functions that may be impacted by nutritional status or interventions.

The text has been changed as follow:

P10 L 267-

Having generated functional profiles of the bacterial communities of each child at baseline and after six months of nutritional intervention, we were interested in establishing a functional hypothesis for the changes observed. We used PICRUST2 software to predict the functional potential of faecal bacterial communities based on marker gene sequencing profiles [22]. We first compared the differential KEGG pathways between children who differed in their nutritional status, micronutrients, inflammatory status and parasitic infection at the baseline.

10. Line 223: "gastrointestinal inflammation, which has never been significantly associated with faecal bacterial composition in all previous analyses..." These types of statements are too definitive and in this case, untrue. See the literature from Andreas Baumler's lab on inflammatory pathogens and microbiota configuration, plus a broad immunological literature.

We agree with the reviewer that this sentence was leading to confusion. Indeed, were not mentioning the literature, but only the results observed in the present study. Following the recommendation of another reviewer, we used a cut-off for PICRUST analysis to get more readable figures and focus only on the strongest association, and gastrointestinal inflammation was no longer associated with any function. Thus, we deleted the sentence and added a paragraph in the discussion section as follow:

P15 L408-

Gastrointestinal inflammation has been linked with changes in the structure of the microbiota [47] In the present study, less than 3% of the children presented a gastrointestinal inflammation, which may explain the lack of association with specific features observed.

11. Lines 273-275: The authors refer to the work in gnotobiotic animals describing micronutrient-microbiota interactions as not pertinent. I agree that more human studies relating micronutrient status to the microbiota are necessary, but there are ethical considerations that make the ideas discussed in the animal study literature very important.

We totally agree with the remark of the reviewer since gnotobiotic animals are the only model allowing to differentiate the role of gut microbes from the host. The idea was not to minimise the pertinence of the use of gnotobiotic animals, but more to underline the discrepancies with the human model. We modified the sentence to make it more explicit in the text as follow:

P13 L338-

The regulation of zinc absorption is less complex than the regulation of iron [34]. Zinc deficiency often leads to diarrhoea, and many studies have focussed on the effect of supplementation at different doses in the poultry and pig industries, which have been shown to cause compositional and functional changes of the gut microbiota [35]. One original study in non-pathogenic conditions of initially germ-free mice inoculated with a mix of different bacteria from the human gut, showed almost no effect of zinc status on bacterial features [36]. This result is in line with the results of the present study, as we observed no within- or between-individual variations in the faecal microbiota of children with different zinc status, pointing to no covariation of microbiota with zinc status. This may be due to the small number of children with normal zinc status in our study. Yet a recent study of 67 schoolchildren in China showed that, despite the absence of beta-diversity difference between control and zinc deficient children, *Coprobacter*, *Acetivibrio*, *Paraprevotella* and *Clostridium_XI* were more abundant in the zinc deficient group [37].

12. Line 314: “first to describe” – this does not constitute impact.

We modified the sentence to present the impact of the study as follow:

P15 L415-

Our results contribute novel information to the description of the faecal microbiota of a cohort of Cambodian schoolchildren, ...

Reviewer #2 (Remarks to the Author):

Summary

This study is a secondary analysis of data and specimens originating from the FORISCA trial (2012-13), a double-blinded, cluster-randomized, placebo-controlled trial including 9500 children that tested the impact of rice fortified with vitamins and minerals on nutritional status and anthropometry of children aged 6–14 years in Cambodia. The trial found nutritional improvements, but a perplexing increase in hookworm infections and increased systemic and intestinal inflammation due to the intervention. This secondary analysis sought to understand the microbiota-mediated mechanisms for the FORISCA trial results. The authors examined microbiota composition in 380 children enrolled in FORISCA at baseline and again at the end (endline) of the 6-month fortified rice intervention. Associations between microbiota composition and inferred (PICRUST) microbiota function, based on 16S rRNA sequencing of 760 fecal samples, and the following factors: age, sex, nutritional status (underweight, stunting), micronutrient deficiencies, inflammation (systemic, gut), and parasitic infection, were evaluated. The microbiota was taxonomically characterized at the genus or higher. Child fecal microbiota exhibited a higher than expected abundance of Lactobacillaceae. There were

associations between the microbiota and micronutrient deficiencies and the fortified rice trial intervention.

Major comments

P2 There are a number of papers that investigate the relationship between the microbiota and malnutrition/undernutrition in children from LMICs, especially those papers produced by the Jeff Gordon and Indi Trehan groups at Washington University and AfriBiota, among others.

We agree that the introduction section would benefit from a more complete description of the relationship between microbiota and undernutrition. The introduction is now as follow:

P3 L70-

While a large body of literature reports associations between the human microbiome and obesity, even if the results are inconsistent [5], data on bacterial microbiota in children with acute malnutrition are relatively scarce but once again underline the importance of gut microbiota [6]. Indeed, severe and acute malnutrition is associated with significant relative microbiota immaturity [7,8]. The use of gnotobiotic mice also identified the gut microbiome as a causal factor in severe acute malnutrition, in addition to insufficient nutrient intake [9]. Significantly, the use of microbiota-targeted complementary food was more efficient than ready-to-use supplementary foods in increasing weight in moderately malnourished Bangladeshi children [10].

P5 The prevalence of iron and zinc deficiencies were 51% and 89%, in the sample of 380 children, but the population-wide zinc deficiency prevalence estimate of 50% was much lower compared to the sample. Vitamin A deficiency was also lower. Can the authors explain the significant differences in micronutrient deficiencies between the population and their sample? Was this likely to be random or systematic sampling error in the selection of subjects from the larger trial into this study?

The numbers presented for the country level population in the introduction section are related to children under 5 years of age. The present study was done on schoolchildren, where no national data are available in Cambodia. (P4 L82-).

Concerning the FORISCA study, the proportion of children with zinc and vitamin A deficiency is similar in all the studies (Kuong et al 2019, Perignon et al 2016, Fiorentino et al 2018). There was a typo in the text, which is now corrected as follow:

P4 L84

Micronutrient deficiencies were highly prevalent with $\geq 80\%$ of children found to be deficient in zinc [14].

Kuong K, Tor P, Perignon M, et al. Multi-Micronutrient Fortified Rice Improved Serum Zinc and Folate Concentrations of Cambodian School Children. A Double-Blinded Cluster-Randomized Controlled Trial. *Nutrients*. 2019;11:2843

Perignon M, Fiorentino M, Kuong K, et al. Impact of Multi-Micronutrient Fortified Rice on Hemoglobin, Iron and Vitamin A Status of Cambodian Schoolchildren: a Double-Blind Cluster-Randomized Controlled Trial. *Nutrients*. 2016;8. doi: 10.3390/nu8010029

Fiorentino M, Perignon M, Kuong K, et al. Effect of multi-micronutrient-fortified rice on cognitive performance depends on premix composition and cognitive function tested: results of an effectiveness study in Cambodian schoolchildren. *Public Health Nutr.* 2018;21:816–27.

P5 Suggest moving the short paragraph (“The impact of the intervention ...”) describing the impact of the intervention on micronutrient deficiencies to the introduction, and add results for stunting, which seem to be missing from the FORISCA trial summary in the introduction.

As required by all the three reviewers, we have expanded the sections on the FORISCA trial in the introduction and results sections. The text is now:

P4 L85-

The aim of the ‘Fortified Rice for School Children in Cambodia’ (FORISCA) project was to assess the impact of rice fortified with different levels of vitamins and minerals on nutritional status, development and anthropometry of schoolchildren aged 6–14 years, who received fortified rice or normal rice for six months. The FORISCA trial showed that daily consumption of three types of fortified rice with different micronutrient content provided through a school meal programme improved the status of most micronutrients [12,13], had a small but nevertheless significant impact on cognitive development [14], but simultaneously increased the prevalence of hookworm infection [15].

P5 L 123

This study involved a subset of 380 children randomly selected from the parent FORISCA double-blind cluster randomized controlled trial using multi-micronutrient fortified rice (ClinicalTrials.gov identifier NCT01706419). The characteristics of the participants at baseline are listed in Supplementary Table 1.

P6 L 133-

The 380 schoolchildren had been randomized (with school as cluster) to receive either normal rice (Placebo), or one of the two types of fortified rice: UltraRice Original formulation (UR-original) or UltraRice Improved formulation (UR-improved) for a period of six months. UR-original was fortified with four micronutrients (iron, zinc, vitamin B1 and B9). UR-improved had four additional vitamins (vitamin A, vitamins B3, B6 and B12). Their final composition differed slightly, with UR-original providing slightly more iron and zinc than UR-improved (Supplementary Table 2).

P5 Was there other information about the included children, for example, information on antibiotic or other medication use, non-rice related dietary factors, or underlying illnesses (e.g., diarrhea). As the authors mention in the introduction, these could be important confounders in these observational data stemming from a clinical trial. It would be very useful to present these data, indicate whether these factors were included in the analysis, or add to a limitations section in the discussion.

The mentioned parameters are of course of importance but all children were dewormed at the start of the trial (according to national guidelines). Also, antibiotic use was (and is) very low in this setting. A more detailed information on the inclusion criteria are now in the material and method section as follow:

P 16 L 430-

A random selection was performed of children for whom all data were available at both points where stool samples had been collected for assessment of parasitic infection and gut inflammation status and who had not taken any antibiotics for the past three months. Children were eligible for inclusion in the sub-study if written informed consent was obtained from their parents or caretakers and verbal assent from the participating children prior to enrolment in the study. Exclusion criteria included being less than 6 years old or more than 14 years old, mental or physical disabilities, or severe anaemia (defined as haemoglobin concentration < 70 g/L). Children diagnosed with severe anaemia were provided with multiple micronutrient supplements for two months.

P5 The authors provide an estimate of the average read counts per sample, but there are no details about the range or distribution of read counts (some datasets could be very small or large), the quality of the reads (e.g., how many reads had to be quality filtered), and there is no indication of results for negative/process or positive controls.

Thank you for the comment. The number of reads varied from a minimum of 2,948 to a maximum of 314,227, with an average read count of 39,355. Following sequencing, a total of 44,129,385 reads were generated, and after quality filtering, 67.7% of the sequences were retained, resulting in 29,910,230 reads. The study did not incorporate positive or negative controls. The text reads now:

P 6 L149-

The number of reads varied from 2,948 to 314,227, with an average read count per sample of 39,355. Following sequencing, a total of 44,129,385 reads were generated, and after quality filtering, 67.7% of the sequences were retained, resulting in 29,910,230 reads from 760 faecal samples.

P5 What was the rationale for a 95% threshold for shared microbiota? This is a very high criterion to use. The endline microbiota results for the core microbiota were not shown, was there a reason for this? Did the authors consider whether the “core genera” were the same in each intervention group, especially for the endline sample? It seems the endline core microbiota members could differ due to the intervention received, and this could be important in the context of this report. Alternatively, the core changes could be due to age, and be the same across the trial intervention arms.

Thank you for the comment. As no general consensus on the definition of “core microbiota” is defined, we chose one out of the existing literature which we defined as genera present in 95% of the population.

Salonen A, Salojärvi J, Lahti L, *et al.* The adult intestinal core microbiota is determined by analysis depth and health status. *Clin Microbiol Infect* 2012;**18**:16–20. doi:10.1111/j.1469-0691.2012.03855.x

Falony G, Joossens M, Vieira-Silva S *et al.* Population-level analysis of gut microbiome variation. *Science* 2016;352:560–4. DOI: [10.1126/science.aad3503](https://doi.org/10.1126/science.aad3503)

Huse SM, Ye Y, Zhou Y *et al.* A core human microbiome as viewed through 16S rRNA sequence clusters. *PLoS One* 2012;7: e34242. <https://doi.org/10.1371/journal.pone.0034242>

We initially represented the core microbiota at the endline as a whole, which is not relevant. We wanted to underline the evolution of the microbiota along time, which is done a better way in the figure 3. As a deep investigation of the effect of the intervention on the faecal microbiota is described in the paragraph entitled “Consumption of multiple micronutrient-fortified rice at school for six months modifies the children’s faecal bacterial composition”, we removed this sentence, and kept the detailed analyse presented **P9 L240-**

P6 Could the authors provide a rationale for why so many beta diversity metrics were tested? If the authors believe the Bray-Curtis or Jaccard is a better fit for their data then it would be helpful to explain why and how this helps their interpretation. Otherwise, this may increase, unnecessarily, the possibility of a false positive.

The rationale for using multiple beta diversity metrics is to gain a more comprehensive understanding of the community dissimilarity, potentially reducing the risk of drawing false conclusions based on a single metric that may not fully capture the underlying pattern in the data.

P6 It seems that age is related to differences in microbiota diversity. Since, this is the case, then age should be accounted for when evaluating changes in microbiota diversity related to the different micronutrient deficiencies. If the authors did account for this, then please clarify.

Thank you for your insightful comment. We appreciate your attention to detail and took this remark into consideration. We took the potential confounding effect of age into account when evaluating changes in microbiota diversity related to micronutrient deficiencies by using a linear mixed model that adjusted for both sex and age. The text was modified as follow in the results and material and methods sections:

P9 L 229-

Age and sex were identified as potential confounding variables, as they were associated with differences in beta-diversity (Table 1) and in composition (Supplementary Figure 2). Therefore, the significant features identified by LefSE analysis were integrated in a linear mixed model, with age groups (6-9 and 10-14 years old) and sex included as covariates. The significant taxa identified through this analysis were found to be robust, as they remained significant even after adjusting for these covariates (Supplementary Table 4). Pielou's evenness indices and Shannon index showed lower alpha diversity in children with anaemia or iron deficiency anaemia (Supplementary Table 5). This suggests that bacterial features associated with anaemia in general are strong, while features associated with vitamin A deficiency, stunting, systemic inflammation and parasites exhibited comparatively weaker associations.

P18 L502-

Additionally, a linear mixed model incorporating covariates such as sex and gender was used to further validate the robustness of significant features. Significance was set at $P < 0.05$.

P19 L511-

Following the LefSe analysis, taxa showing significant differences were further tested in a linear mixed model with log-transformed values adjusted for covariates such as sex and gender. P values were corrected for multiple comparison using the Benjamini-Hochberg procedure (q -value < 0.1).

P6 “Pielou’s evenness was significantly different in children with or without anaemia, iron deficiency anaemia, vitamin A deficiency.” The authors state that these values of microbiota alpha diversity are different, but would be informative to state how they differ (e.g., was the alpha diversity higher or lower in younger versus older children). Multiple associations (e.g., diversity measures and micronutrient deficiency results) are described in the manuscript, but often the direction of the association is omitted.

We indicated the direction of the change in the table and also in the text as follow:

P7 L168-

Pielou’s evenness distribution differed significantly in children with or without anaemia, iron deficiency anaemia, vitamin A deficiency: children with deficiency had a lower diversity index than non-deficient children (Table1). When the Shannon index was used, children with or without vitamin A deficiency exhibited different richness (number of species present in each sample) and evenness (species abundance) in the same direction.

P7 L176-

Both Pielou’s evenness and Shannon indexes revealed differences, the older children having a higher diversity index than the younger ones (Table1).

P6 Was PERMANOVA used for the alpha diversity comparisons, it is not clear from the first sentence on the page. It is clear PERMANOVA is used for testing differences in beta diversity in the methods, but that sentence is confusing.

Thank you for the comment. Details were only in the material and method section, we modified the text and the figure legend to be clearer about the application of the two different statistical techniques as follow:

P7 L162-

Differences in microbiota among children grouped according to age, sex, nutritional, micronutrients, inflammatory status and parasitic infection were evaluated based on their alpha-diversity using Kruskal Wallis statistical analysis and on their beta-diversity using Permutational Multivariate Analysis of Variance (PERMANOVA) analysis (Table1).

P30

Table 1: Baseline alpha- and beta-diversity (pairwise distance) metrics of the 380 children grouped according to age, sex, nutritional micronutrient and inflammatory status and parasitic infection.

Differences in alpha-diversity were estimated using a one-way Kruskal Wallis test. The direction of the change is indicated by “<”. Differences in beta-diversity were estimated using PERMANOVA one-way analysis with 999 permutations. The number of samples in each subcategory is indicated in brackets. For all alpha diversity measures, the median \pm standard deviation is presented alongside the number of participants (n, median \pm standard deviation).

P6 Add reference to Figure 3 -- related to differences in beta diversity by age. The PCoA plot for the age groups is very different. Were the baseline and endline samples run separately, in batches, or all mixed together? Is it possible that some of the divergence is due to differences in run effects?

The figure 3 is now cited in the text and results section was clarified. The samples were run all together several times. So, the divergence is not due to differences in run effects. A sentence has been added in the results and material and method section as follow:

P10 L242-

To establish the effect of the intervention strategy on the faecal bacterial composition, we chose first a PCoA representation of Bray-Curtis dissimilarity index of all the children at baseline (n = 380) and at the end of the 6 months intervention (n = 380) (Figure 3).

P18 L 487-

All the samples were run together in different batches to avoid a difference in run effects.

P7 The authors have already established that there are differences in microbiota composition by age and sex. Lefse does not allow for the inclusion of confounding variables such as age and sex, consider reporting the Maaslin results to account for these factors.

Thank you for your insightful comment, which was shared by the other reviewers. We took the potential confounding effect of age into account when evaluating changes in microbiota diversity related to micronutrient deficiencies by using a linear mixed model that adjusted for both sex and age. The text was modified as follow in the results and material and methods sections:

P9 L229-

Age and sex were identified as potential confounding variables, as they were associated with differences in beta-diversity (Table 1) and in composition (Supplementary Figure 2). Therefore, the significant features identified by LefSE analysis were integrated in a linear mixed model, with age groups (6-9 and 10-14 years old) and sex included as covariates. The significant taxa identified through this analysis were found to be robust, as they remained significant even after adjusting for these covariates (Supplementary Table 4). Pielou's evenness indices and Shannon index showed lower alpha diversity in children with anaemia or iron deficiency anaemia (Supplementary Table 5). This suggests that bacterial features associated with anaemia in general are strong, while features associated with vitamin A deficiency, stunting, systemic inflammation and parasites exhibited comparatively weaker associations.

P18 L502-

Additionally, a linear mixed model incorporating covariates such as sex and gender was used to further validate the robustness of significant features. Significance was set at $P < 0.05$.

P19 L511-

Following the LefSe analysis, taxa showing significant differences were further tested in a linear mixed model with log-transformed values adjusted for covariates such as sex and gender. P values were corrected for multiple comparison using the Benjamini-Hochberg procedure (q -value < 0.1).

P8 Based on type of enrichment made to the rice, did the authors have an a priori idea of what functional pathways would be impacted by the enriched rice formulations? The interpretation of links between microbiota functions and fortified rice could also be strengthened in the discussion.

Thank you for the interesting comment. indeed, the two enriched rice differ by the type and amount of micronutrients and different impacts were expected. To strengthen the discussion, we modified the text as follow:

P14 L362-

In the two rice treatments, the type of enrichment was different, UR-original being enriched mainly in iron and zinc, while UR-improved contained additional vitamin A but less iron. Thus, the impact of the two enriched rice formulations on the composition and the predicted functions of the bacterial faecal microbiota were expected to be different. Indeed, the bacterial faecal composition of children who received UR-original was more affected than that of children who received UR-improved. It has been reported in the literature that high iron intakes expose the gut microbiota to iron overload, which has repeatedly been shown to increase enterobacteria, including enteropathogenic *Escherichia coli* and to produce adverse effect including diarrhoea, especially in infants and young children [19]. However, other studies reported no effect of iron consumption on the incidence of diarrhoea [19]. The collection of data on morbidity in the Placebo and intervention groups, specifically diarrheal episodes, would have been a plus.

The association between bacterial abundances and the consumption of UR-original underlines the fact that bacteria belonging to the family *Lachnospiraceae* are among the most sensitive to the consumption of micronutrients. The *Lachnospiraceae* family was enriched and associated with baseline absence of vitamin A deficiency, anaemia and iron deficient anaemia. Members of the *Lachnospiraceae* family are known to produce metabolites that are beneficial for the host, but its abundance has been shown to increase in subjects with different diseases, mainly those linked to obesity [40]. In our study, which included no obese children, *Lachnospiraceae* could be a good candidate as a biomarker for absence of vitamin A deficiency and iron deficiency anaemia. To better understand the role of this family in human nutritional status, it would be useful to assess this genus in epidemiological studies characterising the micronutrient status in people with different nutritional statuses.

We expected a change in the functions involved in iron and zinc intake by bacteria with the consumption of the two rice. In addition, we expected a change in vitamin A intake and use between baseline and the end of the intervention We expected a difference in vitamin A intake by bacteria in faecal samples collected from children consuming UR-improved. This was not the case since the differences in predicted functions were surprisingly only related to lipid metabolism. Only a few data are available concerning the bacterial functional aspects of the effect of iron intakes on gut microbiota. For example, it has been shown in an in vitro model of child microbiota that high iron medium reduced not only the butyrate concentration but also the concentration and expression of the gene *butCoAT*, which codes for the butyryl-CoA:acetate CoA-transferase involved in the last step of butyrate production [41]. Different studies using in vitro or animal models associated a modification in the concentration of short chain fatty acids with iron and zinc intakes [42–44]. Another study of the effect of iron supplementation in mice treated with antibiotics showed that iron overload increased the proportion of predicted genes involved in primary bile acid biosynthesis, nitrogen and cyanoamino acid metabolism and pentose phosphate pathways, which was described by the authors as limiting the high oxidative effect of iron [45]. In the present study, we observed no increase in any of these functions.

P8 For the PICRUST analysis, and supplemental results, consider restricting the results by significance or highest effect size. This will shorten the Supplemental Figure 3 (especially A and G) and make it readable. Was PICRUST only performed on the 16S rRNA data from the

endline sample? Would it be helpful to compare the functions that are found in both time-points to separate out common functions that persist over time, and then focus on those that differ, especially in the comparison across intervention groups? This might reduce the number of variables to focus on.

We appreciate your feedback and have incorporated it into our analysis. To strengthen our results, we applied effect size cutoffs to filter the statistically significant features identified in PICRUSt. For the intervention comparison between baseline and endline measurements, as well as for the comparison between the intervention groups, we removed features with effect sizes less than 0.2.

We modified the results and the figure caption (P 37 supplementary figure 3) in accordance as follow:

P10 L267-

Having generated functional profiles of the bacterial communities of each child at baseline and after six months of nutritional intervention, we were interested in establishing a functional hypothesis for the changes observed. We used PICRUSt2 software to predict the functional potential of faecal bacterial communities based on marker gene sequencing profiles [22]. We first compared the differential KEGG pathways between children who differed in their nutritional status, micronutrients, inflammatory status and parasitic infection at the baseline.

The enzymatic pathways were seen to differ depending on the type of intervention as well as on anaemic status, iron deficient anaemia status, vitamin A and zinc status. Children without anaemia had more functions involved in amino acid metabolism than non-anaemic children (Supplementary Figure 3A). A similar trend was found for iron deficient anaemic children (Supplementary Figure 3B). Vitamin A deficiency was associated with a reduced abundance of three enzymatic pathways (Supplementary Figure 3C). Zinc deficiency was mainly associated with increased abundance of the formaldehyde oxidation pathway (Supplementary Figure 3D).

The difference in the mean proportion was never more than 0.18% for the functions associated with the different metadata cited above. On the contrary, the change in functions over time was stronger and a cut-off effect size > 0.2 was applied to focus on the strongest associations. The functions of the Placebo group evolved over time with an increase in the number of functions involved in fatty acid biosynthesis (Supplementary Figure 3E). At baseline, higher number of functions involved in peptidoglycan biosynthesis and maturation as well as functions related to bacterial metabolism (fermentation, sucrose degradation, Bifidobacterium shunt (fructose phosphate pathway) use of acetylene and amino acid synthesis) was observed. No differences between Placebo and UR-improved groups were observed (data not shown). Only functions involved in fatty acid metabolism were more frequent in the Placebo and UR-original groups (Supplementary Figure 3F). Stearate biosynthesis was more present in the UR-original group than in the UR-improved group (Supplementary Figure 3G).

P37

Supplementary Figure 3.

Differential KEGG pathways in A) Anaemia (anaemic (n=76) and non-anaemic (n=303)), B) Iron deficiency anaemia (yes (n=47) and no (n=332)), C) Vitamin A deficiency (yes (n=30) and no (n=349)), D) Zinc deficiency (yes (n=230) and no (n=35), E) Baseline and endline Placebo (n=130) with effect size of > 0.2 , F) Endline Placebo (n=130) and UR original (n=115) with effect size of > 0.2 , G) Endline UR original (n=115) and UR improved (n=122) with effect size

of > 0.2 . Group significance was measured using a two-tailed Welch's t-test. The bar plot on the left shows the mean proportion of each group. The dot plot on the right shows differences in the mean proportion.

P12 Could the authors comment on whether the taxa or functions associated with inflammation and parasitic infection overlap?

Thank you for your question. There was no taxa or functions (with the new cut-offs) associated with inflammation in our study, while some features were associated with parasitic infection. So, there were no overlap.

P13 The authors should add a section in the methods on how the stool specimens were collected. They should also comment on whether the samples were stored stably and if they were subject to any freeze-thaw cycles.

The details on the stool specimen collection, transport and storage are now in P16 L443-

Anthropometric measurements were taken, and blood and faecal samples were collected at baseline and at the end of the 6-month intervention. Stool samples were collected by providing school children with a sample collection container and requesting them to return the next day to school with a fresh stool sample. Samples were transported in cool boxes containing cool packs (4°C) to the capital Phnom Penh within 4 hours of collection, and further prepared and stored at -20°C until analysis in Phnom Penh, or transported on dry ice to site of the analysis.

P13. Please provide a reference(s) for the weight and height “standard” measurement procedures.

We included a more precise description of weight and height standard measurement as follow:

P17 L 452

The weight and height of the children were measured without footwear and wearing minimum clothing using standard procedures. Weight was measured once to the nearest 100g (model 881U scale; Seca, Hamburg, Germany). The accuracy of the scales was checked every day using a set of two calibration weights. Height was measured twice to the nearest 0.1 cm on a wooden stadiometer (UNICEF-Cambodia) and mean values were used. When differences between two measurements of height in the same child exceeded 0.5 cm, the measurements were repeated. Height-for-age Z-score (HAZ) and BMI-for-age Z-score (BAZ) were calculated according to the WHO 2006 reference [48]. Underweight and stunting were defined as $\text{BAZ} < -2$ and $\text{HAZ} < -2$, respectively

P14. The logistic mixed models methods and analysis section refers to Maaslin, please make that clear in the methods. Provide a cut-off for interpreting coefficients, such as effect size. Again, this will cut down on the variables for interpretation.

Thank you for your comment. The logistic mixed model was used to assess the impact of the intervention on **micronutrient status** and presented in Material and Method (P17 L 474-) and Supplementary Table 3.

The linear mixed model using MaAsLin2 R package was used to investigate the effect of intervention on **microbiome composition** at the genus level. The following text was added in the material and methods to clarify this point.

P19 L517-

We conducted a multivariate association analysis to assess the impact of the intervention on microbiome composition at the genus level. In this analysis, a linear mixed model effect analysis was performed using the MaAsLin2 R package [62]. Only bacterial genera present in at least 10% of the participants in the study were included in the analysis. Subsequently, the bacterial genera underwent total sum scale (TSS) transformation without normalization for the analysis. The intervention group and time (baseline and endline) were used as a fixed effect while subjects were used as a random effect. A q-value threshold of 0.25 was applied to determine statistical significance. Data supporting the findings of this study and script used for data analysis are accessible at <https://github.com/yoh-s/FORISCA.git>.

P14 Add a section on limitations to the discussion.

Thank you for the comment, as we underlined some limitation in the discussion section such as: P13 L346: “This may be due to the small number of children with normal zinc status in our study.”, we chose to add the additional limitation in the different parts to the discussion as follow:

P13 L 358

One limitation of the present study was the use of multiple micronutrient-enriched rice with different micronutrient profiles as well as different amounts of micronutrients, which make it difficult to attribute specific effect to a specific vitamin or mineral.

P14 L371-

The collection of data on morbidity in the Placebo and intervention groups, specifically diarrheal episodes, would have been a plus.

P15 L399-

Whole metagenome shotgun sequencing would have been useful to examine functional profiles in depth.

P15 “functional potential of the 16S rRNA-predicted taxa identified in the samples.”

Done

Figure 2B. Not clear what yes/no refers to in the figure, please add iron deficiency to the graph labels directly. Same for Supplemental Figure 2A.

Done

Table 1. The legend should be clarified; PERMANOVA is used to test differences in beta diversity, but not alpha diversity.

Thank you for the comment. Details were only in the material and method section, we modified the text and the figure legend to be clearer about the application of the two different statistical techniques as follow:

P7 L 162-

Differences in microbiota among children grouped according to age, sex, nutritional, micronutrients, inflammatory status and parasitic infection were evaluated based on their alpha-diversity using (Kruskal Wallis statistical analysis and, on their beta-diversity using Permutational Multivariate Analysis of Variance (PERMANOVA) analysis (Table 1).

P30

Table 1: Baseline alpha- (pairwise difference) and beta- diversity (pairwise distance) metrics of the 380 children grouped according to age, sex, nutritional micronutrient and inflammatory status and parasitic infection.

Differences in alpha-diversity were estimated using a one-way Kruskal Wallis test. The direction of the change is indicated by “<”. Differences in beta-diversity were estimated using PERMANOVA one-way analysis with 999 permutations. The number of samples in each subcategory is indicated in brackets. For all alpha diversity measures, the median \pm standard deviation is presented alongside the number of participants (n, median \pm standard deviation).

Table 1. Shannon index and Evenness values should be presented for each group so the reader can see the magnitude of the differences in the values

We indicated the Shannon index and Evenness values (median plus or minus standard deviation) as well as the direction of the change in the Table 1 and the corresponding text as follow:

P7 L168-

Pielou’s evenness distribution differed significantly in children with or without anaemia, iron deficiency anaemia, vitamin A deficiency: children with deficiency had a lower diversity index than non-deficient children (Table 1). When the Shannon index was used, children with or without vitamin A deficiency exhibited different richness (number of species present in each sample) and evenness (species abundance) in the same direction.

P7 L176-

Both Pielou’s evenness and Shannon indexes revealed differences: the older children had a higher diversity index than the younger ones (Table 1).

Minor comments

The first sentence of the introduction can be removed.

Done

P1 repetitively -> repeatedly

Done

P1 authors -> investigators

Done

P4 “39,355 read counts per sample”

Done

P8 Revise linear to logistic mixed effect model.

We utilized logistic regression to investigate the impact of the intervention on binary outcomes related to **micronutrient status** (Material and Method (P17 L 474-) and Supplementary Table 3).

Simultaneously, a linear mixed model was applied to analyse the intervention's effect on **microbiome composition**, accounting for relevant covariates (P19 L518-).

P12 revise confusing -> confounding

Done

There is a figure 5 legend shown on page 35, but no figure.

Sorry, it was a misprint, now deleted.

Reviewer #3 (Remarks to the Author):

Title: Faecal microbiota of schoolchildren is associated with nutritional status, micronutrient status and markers of inflammation. (NCOMMS-22-45889-T)

Summary: This paper sought to describe the gut microbiota of Cambodian school children and investigate associations with age, sex, nutritional status, inflammation, and parasitic infection.

Reviewer's Comments:

Validity:

1. This manuscript did not have and major flaws prohibiting its publication.

Originality & significance:

1. The results presented are of interest to researchers in nutrition and gut microbiome in children.

2. Cambodia is an understudied setting for gut microbiome research, making this study a valuable addition to the literature.

Data & methodology:

1. What iron formulation was used to fortify the rice?

Ferrous glycinat was used for both versions. It is now added in the table legend P39.

2. Lines 322: What was the inclusion and exclusion criteria for the parent trial? Were there any additional inclusion or exclusion criteria for being included in the substudy? Was antibiotic use taken into account?

Thank you for the comment. We modified the material and methods adding the missing information as follow:

P16 L426-

This present study focusses on a subset of 380 children who participated in the FORISCA project, which was a large, double-blind, cluster randomized, placebo-controlled trial on the impact of fortified rice on the health and cognitive performance of 9,500 schoolchildren. The study was conducted in Cambodia in November 2012 and July 2013, (ClinicalTrials.gov (Identifier: NCT01706419)). A random selection was performed of children for whom all data were available at both points where stool samples had been collected for assessment of parasitic infection and gut inflammation status and who had not taken any antibiotics for the past three months. Children were eligible for inclusion in the sub-study if written informed consent was obtained from their parents or caretakers and verbal assent from the participating children prior to enrolment in the study. Exclusion criteria included being less than 6 years old or more than 14 years old, mental or physical disabilities, or severe anaemia (defined as haemoglobin concentration < 70 g/L). Children diagnosed with severe anaemia were provided with multiple micronutrient supplements for two months. Six schools in Kampong Speu province that were participating in the United Nations World Food Programme school meal programme were randomly selected for the present trial and placed in three intervention groups named UltraRice Original formulation (UR-original), UltraRice Improved formulation (UR-improved) and Normal rice (Placebo) (supplementary Table 2) [12].

3. Lines 338-341: What equipment were used for measuring weight and height?

The measurement of weight and height is now added in the material and methods as follow:

P17 L452-

The weight and height of the children were measured without footwear and wearing minimum clothing using standard procedures. Weight was measured once to the nearest 100g (model 881U scale; Seca, Hamburg, Germany). The accuracy of the scales was checked every day using a set of two calibration weights. Height was measured twice to the nearest 0.1 cm on a wooden stadiometer (UNICEF-Cambodia) and mean values were used. When differences between two measurements of height in the same child exceeded 0.5 cm, the measurements were repeated.

4. Lines 333-335: Please give more details on the fecal sample collection. What kit was or method was used? Were participants trained on sample collection?

We have provided more details in the text now:

P16 L 443-

Anthropometric measurements were taken, and blood and faecal samples were collected at baseline and at the end of the 6-month intervention. Stool samples were collected by providing school children with a sample collection container and requesting them to return the next day to school with a fresh stool sample. Samples were transported in cool boxes containing cool packs (4°C) to the capital Phnom Penh within 4 hours of collection, and further prepared and stored to the place of analysis and stored at -20 °C until analysis in Phnom Penh, or transported on dry ice to site of the analysis.

5. Line 345-348: What methods were used for the nutritional assays? Only the assay for zinc is described.

Thank you for the comment. Precisions were added in the text as follow:

P17 L464-

Iron status (ferritin and soluble transferrin Receptor), vitamin A status (retinol-binding protein) and biomarkers of inflammation (C-reactive protein (CRP) and α 1-acid-glycoprotein (AGP)) were determined at VitMin laboratory (Willstaett, in Germany). All these proteins were measured using a sandwich enzyme-linked immunosorbent assay (ELISA) technique [49].

P17 L469-

Faecal calprotectin was measured (Calpro AS, NorwayELISA) to estimate the gut inflammation [51].

6. Line 208: Please reconsider the use of PICRUSt, as it was not validated for children and may not be appropriate.

PICRUSt has inherent quality control feature, according to that the largest single factor contributing to metagenome prediction accuracy for a given sample was the extent to which organisms from that sample had their genomes sequenced. Therefore, the weighted Nearest Sequenced Taxon Index (weighted NSTI) score were developed to summarize the extent to which microorganisms in a given sample are related to sequenced genomes.

The weighted nearest-sequenced taxon index (weighted NSTI) was used to evaluate the average distance for the ASVs in a given sample to a reference bacterial genome, and higher scores (> 0.15) might suggest the few related referenced with low prediction quality. In our sample weighted NSTI ranged from 0.005 – 0.15 for 742 samples out of 759 (98% of the samples), which shows for the majority of the samples ASVs had good prediction quality. We have attached the Weighted nearest-sequenced taxon index for metagenome as a Supplementary Table 4.

7. Lines 364-380: Were chloroplast and mitochondrial sequences filtered out? Please add.

We indeed removed the chloroplast and mitochondrial sequences. Therefore, the following sentence is added in P18 L497-

Chloroplast and mitochondria sequences were removed.

8. Were there any adverse effects? Particularly in the iron groups? (See Clarity & Context below)

This is one limitation of the study. A paragraph has been added to present the corresponding literature on the adverse effects of iron consumption and the limitation of the study as follow:

P14 L 362-

In the two rice treatments, the type of enrichment was different, UR-original being enriched mainly in iron and zinc, while UR-improved contained additional vitamin A but less iron. Thus, the impact of the two enriched rice formulations on the composition and the predicted functions of the bacterial faecal microbiota were expected to be different. Indeed, the bacterial faecal composition of children who received UR-original was more affected than that of children who received UR-improved. It has been reported in the literature that high iron intakes expose the gut microbiota to iron overload, which has repeatedly been shown to increase enterobacteria, including enteropathogenic Escherichia coli and to produce adverse effect including diarrhoea, especially in infants and young children [19]. However, other studies reported no effect of iron

consumption on the incidence of diarrhoea [19]. The collection of data on morbidity in the Placebo and intervention groups, specifically diarrheal episodes, would have been a plus.

Appropriate use of statistics and treatment of uncertainties:

1. No comments

Suggested improvements:

1. Introduction: Suggest shortening this with the following outline: Paragraph I: Defining the gut microbiota and its roles; II: What affects the gut microbiota and what is currently known about undernutrition and the gut microbiota in children (see reviews by Million et al. 2017, Barratt et al. 2022, Robertson 2020); III: Parent trial (FORISCA) details; IV. The gap and the objective of your study.

Thank you for your remark. We reorganised and completed the introduction following the recommendations from the three reviewers. The additional information requested are now in the introduction section. Nevertheless, we did not shorten it since in our opinion the sentences concerning the individual variation of the faecal microbiota, the populations less studies and the use of bacterial biomarkers is of importance in our study. Especially thanks to the additional analysis performed to answer the questions of the reviewers, which are really improving the article.

Citations were incorporated and a “gap” section is now added before the objectives as follow:

P3 L 70-

While a large body of literature reports associations between the human microbiome and obesity, even if the results are inconsistent [5], data on bacterial microbiota in children with acute malnutrition are relatively scarce but once again underline the importance of gut microbiota [6]. Indeed, severe and acute malnutrition is associated with significant relative microbiota immaturity [7,8]. The use of gnotobiotic mice also identified the gut microbiome as a causal factor in severe acute malnutrition, in addition to insufficient nutrient intake [9]. Significantly, the use of microbiota-targeted complementary food was more efficient than ready-to-use supplementary foods into increasing weight in moderately malnourished Bangladeshi children [10].

P4 L95-

Numerous studies have investigated the role of gut microbiota in relation to micronutrient status. Indeed, early work comparing germ-free and conventional animals showed a small but crucial role of the gut microbiota on the availability of different B-vitamins (B5, B8, B9, B12 but not B1) in animals fed a deficient diet [17]. Also zinc requirements were also found to be reduced in the germ-free state, which was not the case for iron requirements [17].

P4 L 103-

Studying the faecal bacterial composition of underexplored populations such as schoolchildren in non-Western countries, and its relationship with nutritional status is crucial to advance existing knowledge. Furthermore, double-blinded cluster -randomized controlled trials that investigate the effect of nutritional intervention on faecal bacteria composition in a large number of subjects are also rare in the literature [20].

2. Results:

a. Lines 105-111: Please add the full age range of the children.

Done

b. Lines 113-116: “UR-original” and “UR-improved” are not previously defined (perhaps due to the methods section being at the end). Please check and introduce/describe at first mention in the results section. Also, a placebo is mentioned later, and it is unclear if the placebo is one of the above interventions.

A paragraph is now added to define the two rice and placebo as follow:

P6 L 133-

The 380 schoolchildren had been randomized (with school as cluster) to receive either normal rice (Placebo), or one of the two types of fortified rice: UltraRice Original formulation (UR-original) or UltraRice Improved formulation (UR-improved) for a period of six months. UR-original was fortified with four micronutrients (iron, zinc, vitamin B1 and B9). UR-improved had four additional vitamins (vitamin A, vitamins B3, B6 and B12). Their final composition differed slightly, with UR-original providing slightly more iron and zinc than UR-improved (Supplementary Table 2).

c. Line 122: I believe the term “Bacteroidetes”, which is synonymous with Bacteroidota, is more commonly used; please consider using this term instead.

The term “Bacteroidota” is the new term, which is only recently in use. We indicated it in the text as follow:

P6 L154-

Bacteroidota (formerly termed Bacteroidetes),

References:

1. See above

Clarity and context:

1. Considering the intervention of interest is a high-iron / fortified rice, it would be very interesting to understand how your results compare to the results described by the Zimmermann group (Jaeggi et al. 2015, Paganini 2017 and 2019, etc.) and to put this study in the context of iron interventions and the gut microbiome.

Thank you for this comment that was also made by other reviewers. We now added a paragraph section in the discussion to compare the present study to other works for the effect of micronutrient consumption on the composition and functions of the gut microbiota as follow:

P14 L 362-

In the two rice treatments, the type of enrichment was different, UR-original being enriched mainly in iron and zinc, while UR-improved contained additional vitamin A but less iron. Thus, the impact of the two enriched rice formulations on the composition and the predicted functions of the bacterial faecal microbiota were expected to be different. Indeed, the bacterial faecal composition of children who received UR-original was more affected than that of children who received UR-improved. It has been reported in the literature that high iron intakes expose the gut microbiota to iron overload, which has repeatedly been shown to increase enterobacteria,

including enteropathogenic *Escherichia coli* and to produce adverse effect including diarrhoea, especially in infants and young children [19]. However, other studies reported no effect of iron consumption on the incidence of diarrhoea [19]. The collection of data on morbidity in the Placebo and intervention groups, specifically diarrheal episodes, would have been a plus.

The association between bacterial abundances and the consumption of UR-original different interventions underlines the fact that bacteria belonging to the family Lachnospiraceae are among the most sensitive to the consumption of micronutrients. The Lachnospiraceae family was enriched and associated with baseline absence of vitamin A deficiency, anaemia and iron deficient anaemia. Members of the Lachnospiraceae family are known to produce metabolites that are beneficial for the host, but its abundance has been shown to increase in subjects with different diseases, mainly those linked to obesity [40]. In our study, in which included no children were obese children, Lachnospiraceae could be a good candidate as a biomarker for absence of vitamin A deficiency and iron deficiency anaemia. To better understand the role of this family in human nutritional status, it would be interesting useful to assess this genus in epidemiological studies characterising the micronutrient status in people with different nutritional statuses.

We expected a change in the functions involved in iron and zinc intake by bacteria with the consumption of the two rice. In addition, we expected a change in vitamin A intake and use between baseline and the end of the intervention We expected a difference in vitamin A intake by bacteria in faecal samples collected from children consuming UR-improved. This was not the case since the differences in predicted functions were surprisingly only related to lipid metabolism. Only a few data are available concerning the bacterial functional aspects of the effect of iron intakes on gut microbiota. For example, it has been shown in an in vitro model of child microbiota that high iron medium reduced not only the butyrate concentration but also the concentration and expression of the gene *butCoAT*, which codes for the butyryl-CoA:acetate CoA-transferase involved in the last step of butyrate production [41]. Different studies using in vitro or animal models associated a modification in the concentration of short chain fatty acids with iron and zinc intakes [42–44]. Another study of the effect of iron supplementation in mice treated with antibiotics showed that iron overload increased the proportion of predicted genes involved in primary bile acid biosynthesis, nitrogen and cyanoamino acid metabolism and pentose phosphate pathways, which was described by the authors as limiting the high oxidative effect of iron [45]. In the present study, we observed no increase in any of these functions.

REVIEWER COMMENTS

Reviewer #1 (Remarks to the Author):

I appreciate the authors' efforts to respond to all referees' suggestions. Though the narrative and justifications are improved, the manuscript still reads as a list of default analyses rather than a cohesive story. The narrative and methodology themselves (given point 1 below) are described well enough at this point that the take-aways of the paper can be evaluated, but the figures don't make this easy for a reader.

Main points:

- 1) On additional review of the methods, the authors describe a "V4" 16S rRNA sequencing assay, although the primers they cite clearly cover a larger region of the target than the V4 region alone. Since all analyses are based on this nonstandard amplicon, this fundamental analytical choice must be justified - does this amplicon have proven ability to accurately quantitate bacteria across taxonomic groups, etc? Is there a citation for this specific assay or any precedent for it?
- 2) Although the text is much improved in terms of justifying the analyses performed, the figures (particularly Figure 2 and Figure S3) don't make it easy to see the specific effects on the microbiota the authors are attempting to highlight. Figure 4 represents a much more straightforward identification of differentially abundant taxa, although the legend is unclear in terms of what is indicated. Are these genera the "top 50" or are the significant ($q < 0.05$)?
- 3) Multiple referees suggested a more detailed explanation of the trial itself, the Supp Tables included now (2, 3) appear to be the type of information included in the original publication? A summary of the main effects of the trial as pertinent to the current study would have been sufficient.

Reviewer #2 (Remarks to the Author):

The authors should include a disclaimer in the limitations section of the Discussion indicating that the study did not incorporate positive or negative controls.

The authors may have misunderstood the comment regarding the core microbiota analysis. The text on P9 L240 mentions the global changes to the microbiota in response to the intervention (and over time in the subjects); however, the question was whether the core microbiota changed.

The various beta diversity metrics are used capture different characteristics of microbiota composition by subject. The results vary by the metric used, Bray-Curtis vs. Jaccard vs. Unifrac (page 7). If these analyses were intentional, then provide an explanation for why beta-diversity results would differ by iron deficiency or vitamin A deficiency.

Did the authors mean to say “sex and gender” here? “P18 L502- Additionally, a linear mixed model incorporating covariates such as sex and gender was used to further validate the robustness of significant features. Significance was set at $P < 0.05$ ” and on page 19. Did the authors mean age and sex?

This revised sentence does not make sense. “All the samples were run together in different batches to avoid a difference in run effects.” Do the authors mean that baseline and endline were randomized, interspersed and run together to ensure that batches did not contain either all baseline or all endline samples?

Reviewer #3 (Remarks to the Author):

Title: Faecal microbiota of schoolchildren is associated with nutritional status, micronutrient status and markers of inflammation (NCOMMS-22-45889A)

Reviewer’s Comments:

RE: Authors’ responses

1. The authors have addressed all queries accordingly.

Remaining minor comments

1. I am not sure why there is a question mark after “Placebo” in the Supplementary Table 2 title.

2. Supplementary Figure 3: The images are extremely small and zooming it causes blurring. Since it is supplementary and there are fewer space requirements, I suggest breaking this up so that each part (letter) or pair of parts (letters) of the figure has its own page (perhaps horizontal page orientation) to make this actually legible.

Point by point response to the reviewers:

NCOMMS-22-45889A: Faecal microbiota of schoolchildren is associated with nutritional status, micronutrient status and markers of inflammation.

We would like to thank the reviewers for their interesting comments on the manuscript. We have taken into account the reviewer's remarks and tried to address all the points raised to the best of our capabilities. Please find attached our detailed answer to the reviewers' comments.

We hope that these revisions will be suitable to you.

REVIEWER COMMENTS

Reviewer #1 (Remarks to the Author):

I appreciate the authors' efforts to respond to all referees' suggestions. Though the narrative and justifications are improved, the manuscript still reads as a list of default analyses rather than a cohesive story. The narrative and methodology themselves (given point 1 below) are described well enough at this point that the take-aways of the paper can be evaluated, but the figures don't make this easy for a reader.

Main points:

1) On additional review of the methods, the authors describe a "V4" 16S rRNA sequencing assay, although the primers they cite clearly cover a larger region of the target than the V4 region alone. Since all analyses are based on this nonstandard amplicon, this fundamental analytical choice must be justified - does this amplicon have proven ability to accurately quantitate bacteria across taxonomic groups, etc? Is there a citation for this specific assay or any precedent for it?

Thank you for your comment, which allowed to point out a typo and the absence of reference for the primers used. The primers used in the present study are largely used in the literature and they cover a larger region: V3-V5. Indeed, they are:

357F with the following sequence 5'-CCTACGGGAGGCAGCAG-3'

926R, with the following sequence 5'-CCGTCAATTCMTTTRAGT-3'

They are both coming from the initial work of the Human Microbiome Project Consortium [53] and are widely used in the literature. We corrected the text and added the reference as follow:

P18L486

The V3-V5 hypervariable region of bacterial 16SrRNA was sequenced in paired-end mode (2×300 bp) on the MiSeq platform (Illumina, performed by the Research and Testing Laboratory in Lubbock, Texas, US) using primers 357F (5'-CCTACGGGAGGCAGCAG-3') and 926R (5'-CCGTCAATTCMTTTRAGT-3') [53].

[53] The Human Microbiome Project Consortium. Structure, function and diversity of the healthy human microbiome. *Nature*. 2012;486:207–14 <https://doi.org/10.1038/nature11234>.

2) Although the text is much improved in terms of justifying the analyses performed, the figures (particularly Figure 2 and Figure S3) don't make it easy to see the specific effects on the microbiota the authors are attempting to highlight. Figure 4 represents a much more straightforward identification of differentially abundant taxa, although the legend is unclear in terms of what is indicated. Are these genera the "top 50" or are the significant ($q < 0.05$)?

We agree that the figure built from our file are difficult to read, even if the quality of the figures themselves are above 300 dpi, but we improved the quality of some of the figures for the present version. For figures 2 and S3, we chose to present cladograms and not LDA score bar plot, since they present the advantage of phylogenetic tree representation. Indeed, it illustrates taxonomic levels represented by rings, with phyla at the innermost and genera at the outermost ring. Each circle representing bacterial member within that level. For an easier visibility of the figures, we added each of them at the end of the present document P10. We modified the figures legends by adding a few explanatory lines as follow:

P27 and 35

The cladogram representation illustrating taxonomic levels from the innermost phylum ring to the outermost genera ring. Each circle represents a bacterial member within that level. Circles coloured green or red in are significantly associated with the group indicated by the legend of each individual cladogram.

In the figure 4, the genera are the Top 50 differentially abundant genera identified based on the model coefficient values from the MaAsLin2 analysis. In total, as it is written in the text, 136 taxa with FDR-corrected values < 0.05 were identified. We presented the Top 50 taxa with the largest effect sizes for more visibility of the figure. The figure legend was modified as follow:

P29

Figure 4. Statistical analyses of multivariate associations between UR-original ($n = 114$) and UR-Improved ($n = 122$) intervention groups (compared to the Placebo group, $n = 128$) and gut microbiota composition at the genus level of school-aged children. The analysis was performed using MaAsLin2 with adjustments for multiple comparison ($q \text{ value} < 0.05$). Among the 136 genera differentially affected by the intervention, the Top 50 with the largest effect sizes are represented. All detected associations are adjusted for subjects as random effects

3) Multiple referees suggested a more detailed explanation of the trial itself, the Supp Tables included now (2, 3) appear to be the type of information included in the original publication? A summary of the main effects of the trial as pertinent to the current study would have been sufficient.

Thank you for your comment, which allow a clarification of the trial itself. The full trial was on 9500 children and 380 children were randomly selected for the present study. Different articles

with subsamples of this initial large trial were published as indicated in the introduction section P4L88. The supplementary table 2 indicates the micronutrient composition of the rice used in the intervention, and in our opinion, it is useful to have this information accessible easily as a supplementary data.

The supplementary table 3 is the representation of the effect of the intervention on this specific subsample. To make it more clear, we indicated this information in the text as follow:

P4L87

The aim of the ‘Fortified Rice for School Children in Cambodia’ (FORISCA) project was to assess the impact of rice fortified with different levels of vitamins and minerals on nutritional status, development and anthropometry of 9,500 schoolchildren aged 6–14 years, who received fortified rice or normal rice for six months.

P5L124

This study involved a subset of 380 children randomly selected from the parent FORISCA double-blind cluster randomized controlled trial using multi-micronutrient fortified rice on 9,500 children

Reviewer #2 (Remarks to the Author): The authors should include a disclaimer in the limitations section of the Discussion indicating that the study did not incorporate positive or negative controls.

As suggested, we included a sentence in the Discussion section to indicate that no positive control was incorporated. Concerning the negative control, the sequencing was done by Research and Testing Laboratory (Lubbock, Texas, US), which included negative control in their sequencing workflow. We added a sentence in the material and methods to give this information.

P15 L 413

One limitation of the study was that no positive control was incorporated in the sequencing.

P18 L492

A no template control was included at the PCR step and sequenced along with the samples

The authors may have misunderstood the comment regarding the core microbiota analysis. The text on P9 L240 mentions the global changes to the microbiota in response to the intervention (and over time in the subjects); however, the question was whether the core microbiota changed.

Thank you for precisising the question. Indeed, we initially presented the core microbiota at the baseline and mentioned the changes at the endline without adding the endline figure, which is presented below. Nevertheless, after reading your comment, we considered that a unique core

microbiota after six months of three different nutritional interventions was not statistically meaningful. In the same direction, dividing the initial population of 380 children into three smaller groups was also not statistically meaningful. For these two reasons, we chose not to present it in the article, rather than presenting one endline core microbiota for children receiving different nutritional interventions.

The various beta diversity metrics are used capture different characteristics of microbiota composition by subject. The results vary by the metric used, Bray-Curtis vs. Jaccard vs. Unifrac (page 7). If these analyses were intentional, then provide an explanation for why beta-diversity results would differ by iron deficiency or vitamin A deficiency.

The differences observed in beta-diversity results by iron deficiency or vitamin A deficiency can be attributed to the distinct characteristics captured by different metrics such as Bray-Curtis, Jaccard, and Unifrac. Each metric provides unique insights into the composition of the microbiota, influencing the interpretation of diversity patterns. Bray-Curtis distance considers the abundance of taxa present in the samples, focusing on the quantitative differences between microbial communities. Jaccard Distance, in contrast, emphasizes the presence or absence of taxa, disregarding their abundance. Unifrac distances (weighted and unweighted version) measures the phylogenetic distance between microbial communities, thus taxa that are phylogenetically close are considered less distant.

The abundance (significantly different Bray-Curtis distances) of taxa in the faecal samples of children differing by their vitamin A status was different, but not the taxa composition (non-significantly different Jaccard distance). The taxa present in children differing by their vitamin A status were phylogenetically close when the relative abundance was not considered (Unweighted Unifrac distances) but more distant when relative abundance was considered (Weighted Unifrac distances).

The abundance (significantly different Bray-Curtis distance) and presence/absence (significantly different Jaccard distances) of taxa in the children differing by their iron deficiency

status were different. But the taxa were phylogenetically close (Unweighted and Weighted Unifrac distances non-significantly different).

To be more clear in the text, we added a short description of the metrics in the text as follow:

P7L180

Bray-Curtis (abundance of taxa) and Jaccard (presence/absence of taxa) distances differed significantly in children with (n = 193) or without (n = 187) iron deficiency (Table 1). Similarly, Bray-Curtis and weighted Unifrac (different phylogenetic lineages considering the abundance of taxa) distances differed significantly in children with (n = 30) or without (n= 350) vitamin A deficiency.

Furthermore, beta-diversity differed between age groups as shown by their Bray-Curtis, Jaccard and Unweighted Unifrac (phylogenetic lineages of taxa) distances (Table 1). Finally, Jaccard distances differed significantly in boys (n = 211) and girls (n = 169).

Did the authors mean to say “sex and gender” here? “P18 L502- Additionally, a linear mixed model incorporating covariates such as sex and gender was used to further validate the robustness of significant features. Significance was set at $P < 0.05$ ” and on page 19. Did the authors mean age and sex?

Sorry, it was a typo, it should be “age and sex”. It is now corrected in the different place in the article.

This revised sentence does not make sense. “All the samples were run together in different batches to avoid a difference in run effects.” Do the authors mean that baseline and endline were randomized, interspersed and run together to ensure that batches did not contain either all baseline or all endline samples?

The sentence was rephrased as follow:

P18L490

Different intervention groups as well as time points (baseline and endline) were randomized, pooled in four batches of 190 samples and analysed into four runs to ensure that batches did not contain either all baseline or all intervention group samples.

Reviewer #3 (Remarks to the Author):

Title: Faecal microbiota of schoolchildren is associated with nutritional status, micronutrient status and markers of inflammation (NCOMMS-22-45889A)

Reviewer's Comments:

RE: Authors' responses

1. The authors have addressed all queries accordingly.

Remaining minor comments

1. I am not sure why there is a question mark after "Placebo" in the Supplementary Table 2 title.

Corrected

2. Supplementary Figure 3: The images are extremely small and zooming it causes blurring. Since it is supplementary and there are fewer space requirements, I suggest breaking this up so that each part (letter) or pair of parts (letters) of the figure has its own page (perhaps horizontal page orientation) to make this actually legible.

Thank you for your feedback on the supplementary materials. The quality of the figures is high and the lack of quality is due to pdf conversion. We improved the quality of some of the figures for the present version. In addition, to allow you to better see the results, we have adjusted figure size and divided Supplementary Figure 3 into separate pages.

A)

B)

C)

G)

H)

Supplementary Figure 3. Differential KEGG pathways in A) Anaemia (anaemic (n=76) and non-anaemic (n=303)), B) Iron deficiency anaemia (yes (n=47) and no (n=332)), C) Vitamin A deficiency (yes (n=30) and no (n=349)), D) Zinc deficiency (yes (n=230) and no (n=35)), E) Gastrointestinal inflammation with effect size of > 0.1 (yes (n=10) and no (n=316)), F) Baseline and endline Placebo (n=130) with effect size of > 0.5 G) Endline Placebo (n=130) and UR original (n=115) with effect size of > 0.2 H) Endline UR original (n=115) and UR improved (n=122) with effect size of > 0.5. Group significance was measured by two-tailed Welch's t-test.

A

■ Anemia (n = 76)
■ No-anemia (n = 304)

■ a: Prevotella_7
■ b: PrevotellaceaeNK3B31group
■ c: Erysipelatoclostridiaceae
■ d: Limosilactobacillus
■ e: Anaerostipes
■ f: Lachnospira
■ g: Eubacterium_halligroup
■ h: Ruminococcus_torquesgroup
■ i: Lachnospiraceae
■ j: Faecalibacterium
■ k: Klebsiella

B

■ Iron deficiency anaemia (n = 47)
■ No iron deficiency anaemia (n = 333)

■ a: Prevotella_7
■ b: PrevotellaceaeNK3B31group
■ c: HT002
■ d: Lachnospira
■ e: LachnospiraceaeUCG_001
■ f: LachnospiraceaeUCG_010
■ g: Eubacterium_halligroup
■ h: Lachnospiraceae
■ i: Lachnospirales
■ j: Monoglobus
■ k: Monoglobaceae
■ l: Monoglobales
■ m: Faecalibacterium
■ n: Ruminococcus
■ o: Ruminococcaceae
■ p: Oscillospirales

C

D

Figure 2: Differential ASVs between the children grouped according to their nutritional status: A) anaemia, B) iron deficiency, C, vitamin A deficiency, D) stunting. The differences were identified using Linear discriminant analysis Effect Size (LEfSe) analysis with a P-value < 0.05, and a logarithmic LDA (linear discriminant analysis) score of > 2. The cladogram representation illustrating taxonomic levels from the innermost phylum ring to the outermost genera ring. Each circle represents a bacterial member within that level. Circles coloured green or red in are significantly associated with the group indicated by the legend of each individual cladogram. The number of samples in each subcategory is indicated on each graph.

A

Inflammation (n=140)
No inflammation (n=240)

- a: Butyricimonas
- b: Alloprevotella
- c: Unknown
- d: Unknown
- e: Gastranaerophilales
- f: Holdemania
- g: Enterococcus
- h: Enterococcaceae
- i: Lapidilactobacillus
- j: Lentilactobacillus
- k: Unknown
- l: Unknown
- m: RF39
- n: CAG_352
- o: Romboutsia
- p: Peptostreptococcaceae
- q: Enterobacter
- r: Klebsiella
- s: Succinivibrio
- t: Succinivibrionaceae

B

Parasitic infection (n=104)
No parasitic infection (n=276)

- a: Gordonibacter
- b: Desulfovibrionaceae
- c: Kurthia
- d: UCG_004
- e: Holdemanella
- f: Erysipelotrichaceae
- g: Weissella
- h: Sarcina
- i: Fastidiosipila
- j: Hungateiclostridiaceae

C

D

Supplementary Figure 2: Differential ASVs between the children grouped according to A) their systemic inflammation, B) parasite infection, C) age, D) sex. The differences were identified using Linear discriminant analysis Effect Size (LEfSe) analysis with a P-value cut-off of < 0.05 and a logarithmic LDA score of > 2 . The cladogram representation illustrating taxonomic levels from the innermost phylum ring to the outermost genera ring. Each circle represents a bacterial member within that level. Circles coloured green or red in are significantly associated with the group indicated by the legend of each individual cladogram. The number of samples in each subcategory is indicated on each graph.

REVIEWERS' COMMENTS

Reviewer #2 (Remarks to the Author):

Changes to Figure 2 – Revise this sentence in the legend as it is not clear, “Circles coloured green or red in are significantly associated with the group indicated by the legend of each individual cladogram.” Red and green are likely showing enrichment or depletion.

In the authors response they indicate that Figure S3 now includes a cladogram, but in fact it still contains the bar charts; however, they cannot be cladograms because it is not possible to determine which specific taxon is contributing a function(likely multiple are contributing), therefore I think the response is fine. Still figure 3S is very hard to read as a reviewer. Figure 4 is OK.

Point by point response to the reviewers:

NCOMMS-22-45889B: Faecal microbiota of schoolchildren is associated with nutritional status, micronutrient status and markers of inflammation.

We would like to thank the reviewers for their comments on the manuscript. We have considered the reviewer's remarks and tried to address all the points raised to the best of our capabilities. Please find attached our detailed answer to the reviewers' comments.

We hope that these revisions will be suitable to you.

REVIEWERS' COMMENTS

Reviewer #2 (Remarks to the Author):

Changes to Figure 2 – Revise this sentence in the legend as it is not clear, “Circles coloured green or red in are significantly associated with the group indicated by the legend of each individual cladogram.” Red and green are likely showing enrichment or depletion.

Thank you for your comments, which helps us to clarify the figure legend. Red and green are not showing enrichment or depletion, but enrichment for the group of children differing by the status indicated in each individual legend (anaemia, iron deficiency, etc.) We modified the figure legend as follow to make it more clear :

Circles coloured green or red shows significant enrichment of those taxa in the groups indicated by the legend of each individual cladogram.

In the authors response they indicate that Figure S3 now includes a cladogram, but in fact it still contains the bar charts; however, they cannot be cladograms because it is not possible to determine which specific taxon is contributing a function (likely multiple are contributing), therefore I think the response is fine. Still figure 3S is very hard to read as a reviewer. Figure 4 is OK.

Thank you for pointing out this typo. In the previous answer for reviewers, we were referring to Figure S2 and not S3 for the cladograms. The figure S3 contains indeed bar charts showing the differential KEGG pathways in the different groups. The figures were uploaded in a high quality on the online system, to ensure a good readability.